# GraphLLM: Boosting Graph Reasoning Ability of Large Language Model

## Abstract

The advancement of Large Language Models (LLMs) has remarkably pushed the boundaries towards artificial general intelligence (AGI), with their exceptional ability on understanding diverse types of information, including but not limited to images and audio. Despite this progress, a critical gap remains in empowering LLMs to proficiently understand and reason on graph data. Recent studies underscore LLMs' underwhelming performance on fundamental graph reasoning tasks. In this paper, we endeavor to unearth the obstacles that impede LLMs in graph reasoning, pinpointing the common practice of converting graphs into natural language descriptions (*Graph2Text*) as a fundamental bottleneck. To overcome this impediment, we introduce GraphLLM, a pioneering end-to-end approach that synergistically integrates graph learning models with LLMs. This integration equips LLMs with the capability to proficiently interpret and reason on graph data, harnessing the superior expressive power of graph learning models. Our empirical evaluations across four fundamental graph reasoning tasks validate the effectiveness of GraphLLM. The results exhibit a substantial average accuracy enhancement of 54.44%, alongside a noteworthy context reduction of 96.45% across various graph reasoning tasks.[1]

## 1 Introduction

The AI community has witnessed the emergence of powerful pre-trained Large Language Models (LLMs) (Brown et al., 2020; Chowdhery et al., 2022; OpenAI, 2023; Touvron et al., 2023), which leads to the pursuit of the potential realization of Artificial General Intelligence (AGI). Inspired by the fact that an intelligent agent, like the human brain, processes information of diverse types, there is a trend towards empowering LLMs to understand various forms of data, such as audio (Huang et al., 2023) and images (Alayrac et al., 2022). Despite significant strides in interpreting multimodal information (Yin et al., 2023), empowering LLMs to understand graph data remains relatively unexplored. Graphs, which represent entities as nodes and relationships as edges, are ubiquitous in numerous fields, *e.g.* molecular networks, social networks. An intelligent agent is expected to reason with graph data to facilitate many tasks such as drug discovery (Stokes et al., 2020) and chip design (Mirhoseini et al., 2021).

Current efforts have revealed that LLM's performance on some fundamental graph reasoning tasks is (unexpectedly) subpar. As noted by Wang et al. (2023a), even with tailor-made prompts, LLMs muster an accuracy of barely 33.5% when tasked with calculating the shortest path on a graph with up to 20 nodes. Their research also highlighted that fine-tuning OPT-2.7B (Zhang et al., 2022) failed to elicit the graph reasoning ability. Similarly, our experiments indicate that fine-tuning more recent LLaMA2-7B/13B (Touvron et al., 2023) still results in underwhelming performances in several fundamental graph reasoning tasks. This raises an essential question: *What hinders the ability of LLMs on graph reasoning tasks?*

We posit that the key obstacle to LLMs' graph reasoning ability can be attributed to the prevailing practice of *converting graphs into natural language descriptions (Graph2Text)*. A majority of the existing attempts to apply LLMs to graph data, such as the studies by Wang et al. (2023a); Guo et al. (2023); Ye et al. (2023), employ *Graph2Text* strategy to convert graph data into textual descriptions. While the *Graph2Text*-based methodology facilitates direct processing of graph data by

---

[1]Codes and datasets are available at https://anonymous.4open.science/r/GraphLLM-4455.

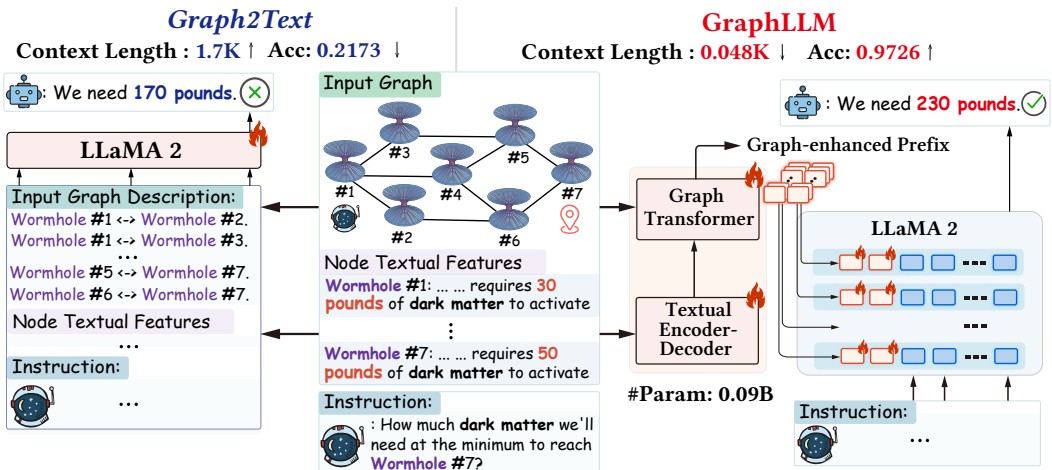

Figure 1: **Demonstration of *Graph2Text* vs. GraphLLM.** The LLM is tasked with computing the minimum quantity of dark matter necessary to transition from the starting wormhole to the ending wormhole, given the connectivity graph and the textual descriptions of each node.

LLMs through textual descriptions, it introduces following inherent shortcomings that curtail the ability of LLMs on graph reasoning tasks:

1. LLMs, when using the *Graph2Text* strategy, are compelled to discern implicit graph structures from sequential text. In contrast to dedicated graph learning models that inherently process graph structures, LLMs may face **difficulties** in learning on graph based on sequential graph descriptions.

2. The *Graph2Text*-based methodology inherently results in a **lengthy context** of graph description, as illustrated in Figure 1. This could pose a challenge for LLMs to identify essential information for graph reasoning tasks from the lengthy contexts (Liu et al., 2023).

To tackle the aforementioned limitations and enhance the ability of LLMs in graph reasoning, we introduce GraphLLM. Contrary to the *Graph2Text* strategy of converting graphs into textual descriptions, GraphLLM's core idea is to synergistically integrate a graph learning module (graph transformer) with the LLM to enhance graph reasoning ability. By synergizing the LLM and the graph transformer, GraphLLM harnesses the strengths of both and offers a more powerful and efficient solution to applying LLMs for graph reasoning tasks. Specifically, GraphLLM possesses the following two key advantages over *Graph2Text*-based methodology:

1. **Collaborative Synergy.** GraphLLM takes an end-to-end approach to integrate graph learning models and LLMs within a single, cohesive system. By synergizing with graph learning models, LLMs can harness its superior expressive power on graph data. Compared to *Graph2Text*-based methodology, GraphLLM achieves an average accuracy improvement from 43.75% to 98.19% on four fundamental graph reasoning tasks.

2. **Context Condensation.** GraphLLM condenses graph information into a concise, fixed-length prefix, thereby circumventing the need of *Graph2Text* strategy to produce lengthy graph descriptions. Compared to *Graph2Text*-based methodology, GraphLLM substantially reduces the context length by 96.45%.

Our experiments on four fundamental graph reasoning tasks covering text substructure counting, maximum triplet sum, shortest path, and bipartite graph matching, demonstrate that GraphLLM boosts the graph reasoning ability of LLM by an average accuracy improvement of 54.44%, while achieving a remarkable context reduction of 96.45% and 3.42x inference acceleration.

## 2 PRELIMINARY

**Definition 2.1.** (`Input` **Graph**) Given an instance of instruction pair (`Input`, `Instruction`, `Response`), the `Input` graph is a set $\mathcal{V}$ of $n$ node $\{\boldsymbol{d}_0, \boldsymbol{d}_1, \ldots, \boldsymbol{d}_{n-1}\}$, where $\boldsymbol{d}_i$ is the textual feature[2] of $i$-th node, with graph structure $\mathcal{E}$ on $\mathcal{V}$. The graph structure $\mathcal{E} : \mathcal{V} \times \mathcal{V} \rightarrow \{0, 1\}$ is defined as follows:

$$\mathcal{E}(\boldsymbol{d}_i, \boldsymbol{d}_j) = \begin{cases} 1, & \text{if there is a relationship between } \boldsymbol{d}_i \text{ and } \boldsymbol{d}_j \\ 0, & \text{otherwise} \end{cases} \tag{1}$$

Thus the `Input` graph $\mathcal{G}$ can be denoted as a tuple $\{\mathcal{V}, \mathcal{E}\}$. *Graph2Text*-based methodology introduces graph description language $\mathcal{A}(\mathcal{V}, \mathcal{E}) \rightarrow$ `TextDescription`.

**Definition 2.2.** (**Fine-tuning on Graph Reasoning Tasks**) Given a pre-trained LLM $\mathcal{M}$ with parameters $\boldsymbol{\theta}$, a dataset of $m$ instruction pairs $\{(\texttt{Input}_i, \texttt{Instruction}_i, \texttt{Response}_i)_{i=0,\ldots,m-1}\}$, where each $\texttt{Input}_i$ is a graph $\mathcal{G}_i = \{\mathcal{V}_i, \mathcal{E}_i\}$, and a task-specific objective function $\mathcal{L}$, the fine-tuning process aims to learn task-specific parameters $\boldsymbol{\theta}^\star$ by minimizing the following loss function:

$$\boldsymbol{\theta}^\star = \arg\min_{\boldsymbol{\theta}'} \sum_{i=0}^{m-1} \mathcal{L}(\mathcal{M}(\mathcal{V}, \mathcal{E}, \texttt{Instruction}; \boldsymbol{\theta}'); \texttt{Response}) \tag{2}$$

where $\mathcal{M}(\ ; \boldsymbol{\theta}')$ represents the output of the fine-tuned LLM $\mathcal{M}$ with parameters $\boldsymbol{\theta}'$. Note that in Eq. (2) the subscripts of $\mathcal{V}, \mathcal{E}$, `Instruction` and `Response` are omitted for clarity.

**Prefix Tuning** Given a pre-trained LLM with an $L$-layer transformer, prefix tuning fixes the original LLM parameters and only prepends $K$ trainable continuous tokens (prefixes) to the keys and values of the attention at every transformer layer. Taking the $l$-th attention layer as an example ($l < L$), prefix vectors $\boldsymbol{P}_l \in \mathbb{R}^{K \times d^{\mathrm{M}}}$ is concatenated with the original keys $\boldsymbol{K}_l \in \mathbb{R}^{* \times d^{\mathrm{M}}}$ and values $\boldsymbol{V}_l \in \mathbb{R}^{* \times d^{\mathrm{M}}}$, where $d^{\mathrm{M}}$ is the dimension of LLM, formulated as:

$$\boldsymbol{K}_l' = [\boldsymbol{P}_l; \boldsymbol{K}_l]; \ \boldsymbol{V}_l' = [\boldsymbol{P}_l; \boldsymbol{V}_l] \in \mathbb{R}^{(K+*) \times d^{\mathrm{M}}} \tag{3}$$

The new prefixed keys $\boldsymbol{K}_l'$ and values $\boldsymbol{V}_l'$ are then subjected to the $l$-th attention layer of LLM. For simplicity, we denote the vanilla attention computation as $\boldsymbol{O}_l = \texttt{Attn}(\boldsymbol{Q}_l, \boldsymbol{K}_l, \boldsymbol{V}_l)$. The computation of attention becomes:

$$\boldsymbol{O}_l = \texttt{Attn}(\boldsymbol{Q}_l, [\boldsymbol{P}_l; \boldsymbol{K}_l], [\boldsymbol{P}_l; \boldsymbol{V}_l]) \tag{4}$$

In vanilla prefix tuning, prefixes are initialized from a trainable parameter tensor $\mathbf{P} \in \mathbb{R}^{L \times K \times d^{\mathrm{M}}}$.

## 3 GRAPHLLM

### 3.1 GENERAL FRAMEWORK OF GRAPHLLM

**Reason on Graphs** Since graphs inherently represent entities and their interrelationships, reasoning on graphs requires simultaneous consideration of both the entities (nodes) and their relationships (edges). Consequently, graph reasoning tasks encompass two sub-objectives: *node understanding* and *structure understanding*. For example, in the context of counting specific substructures within a molecular graph, one must discern the types of atoms from node descriptions (*node understanding*) and recognize the chemical bonds derived from the graph's structure (*structure understanding*). In the proposed GraphLLM, we intentionally devise modules addressing these dual objectives.

As demonstrated in Figure 2, GraphLLM consists of the following three main steps:

---

[2]In this paper, we operate under the assumption that nodes (or entities) can be characterized by textual features, which is applicable to various real-world graphs, such as user profiles in social networks or atom property descriptions in molecular graphs.

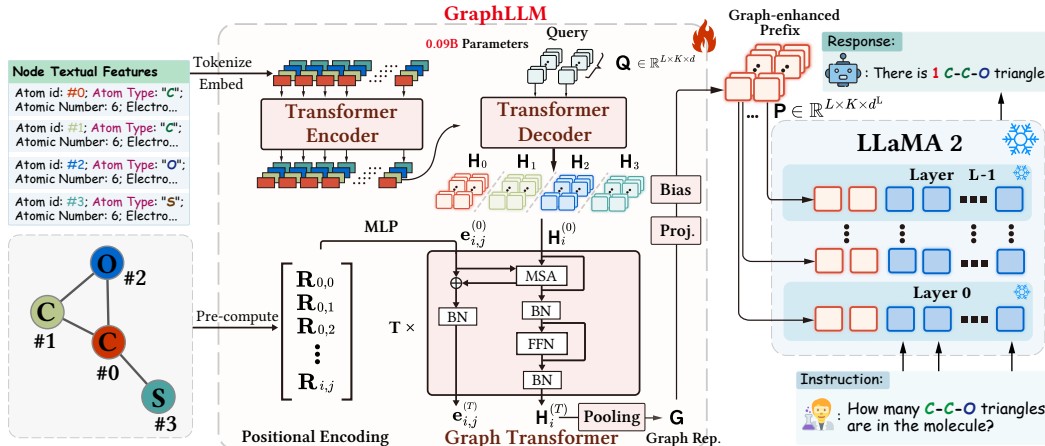

Figure 2: An illustration of reasoning on a toy molecular graph with GraphLLM. The LLM is requested to identify the number of C-C-O triangles in the molecule.

1. *Node Understanding* (§3.2): A textual transformer encoder-decoder is used to extract semantic information crucial to solving graph reasoning tasks from node textual descriptions. The encoder-decoder is newly initialized and updated with the guidance of the pre-trained LLM.

2. *Structure Understanding* (§3.3): A graph transformer is employed to learn on the graph structure by aggregating the node representations obtained from the textual encoder-decoder. In this way, the graph representation produced by the graph transformer can incorporate both node semantic information and graph structure information simultaneously.

3. *Graph-enhanced Prefix Tuning for LLMs* (§3.4): GraphLLM derives the graph-enhanced prefix from the graph representation. During graph-enhanced prefix tuning, the LLM synergizes with the graph transformer by end-to-end fine-tuning, therefore boosting the LLM's capability in conducting graph reasoning tasks with proficiency.

## 3.2 ENCODER-DECODER FOR NODE UNDERSTANDING

The goal of the encoder-decoder is to extract the required information from the nodes based on the specific graph reasoning task. For example, when identifying substructures within molecule, it is necessary to extract atom types from the descriptions of the atoms. For the shortest path task, discerning the cost associated with each node from their descriptions is essential. Therefore, GraphLLM employs a textual transformer encoder-decoder architecture to adaptively extract node information required for graph reason tasks.

Specifically, a textual transformer encoder first applies self-attention to the node description, generating a context vector that captures the semantic meaning pertinent to graph reasoning tasks. Subsequently, a transformer decoder produce the node representation $\mathbf{H}_i$ through the cross-attention between the context vector $c_i$ and the query $\mathbf{Q}$. The query $\mathbf{Q}$ is a newly-initialized trainable embedding. For convenience, we provide a brief overview of the computation process of the encoder-decoder in Eq. (5). Detailed information can be found in Appendix A.1.

$$c_i = \texttt{TransformerEncoder}(\boldsymbol{d}_i \boldsymbol{W}_{\mathrm{D}}) \tag{5a}$$

$$\mathbf{H}_i = \texttt{TransformerDecoder}(\mathbf{Q}; \, c_i) \tag{5b}$$

where $\boldsymbol{d}_i \in \mathbb{R}^{* \times d^{\mathrm{M}}}$ is the embeddings[3] of the textual description of node $i$ (* represents description's length). $\boldsymbol{W}_{\mathrm{D}} \in \mathbb{R}^{d^{\mathrm{M}} \times d}$ is a down-projection matrix to reduce the dimension. $d$ is the dimension of the node understanding encoder-decoder and the structure understanding graph transformer. $c_i \in \mathbb{R}^{* \times d}$ is node $i$'s context vector and $\mathbf{H}_i \in \mathbb{R}^{L \times K \times d}$ is the node $i$' representation. $\mathbf{Q} \in \mathbb{R}^{L \times K \times d}$ is

---

[3]The textual descriptions of the nodes are tokenized and embedded by LLM's tokenizer and frozen embedding table to align with the downstream frozen LLM.

learnable query embedding, where $L$ is the layer number of LLM transformer and $K$ is the length of prefix.

In GraphLLM, we adopt a lightweight transformer encoder-decoder (0.05B parameters for LLaMA 2 7B backbone). In practice, a newly-initialized encoder-decoder can effectively learn to capture node information required for graph reasoning tasks under the guidance of the pre-trained LLM.

## 3.3 GRAPH TRANSFORMER FOR STRUCTURE UNDERSTANDING

Aiming at structure understanding, GraphLLM utilizes a graph transformer to learn from the graph structure. In our framework, the core advantage of the graph transformer over other commonly used graph learning modules (Kipf & Welling, 2017; Veličković et al., 2018) lies in its decoupling of node information and structural information. In the graph transformer, both the positional encoding, which captures the structural information of the graph, and the node representations are independently fed into the transformer blocks and subsequently updated during the learning process. We empirically find that the decoupling of node understanding and structure understanding enhances GraphLLM's graph reasoning ability. The graph transformer primarily consists of two key designs: positional encoding and attention mechanism on graph.

The positional encoding $\boldsymbol{e}_{i,j}$ between node $i$ and node $j$ is initialized using relative random walk probabilities (RRWP) encoding (Ma et al., 2023). Let $\boldsymbol{A}$ be the adjacency matrix of a graph $\{\mathbb{V}, \mathcal{E}\}$ and $\boldsymbol{D}$ be the degree matrix. Define the random walk matrix $\boldsymbol{M} := \boldsymbol{D}^{-1}\boldsymbol{A}$, $\boldsymbol{I}$ the identity matrix. The positional encoding $\boldsymbol{e}_{i,j}$ for each node pair $i, j \in \mathcal{V}$ can be formulated as follows:

$$\boldsymbol{R}_{i,j} = [\boldsymbol{I}_{i,j}, \boldsymbol{M}_{i,j}, \boldsymbol{M}_{i,j}^2, ..., \boldsymbol{M}_{i,j}^{C-1}] \in \mathbb{R}^C \qquad (6a)$$

$$\boldsymbol{e}_{i,j} = \Phi(\boldsymbol{R}_{i,j}) \in \mathbb{R}^d \qquad (6b)$$

in which $C$ is a parameter controlling the maximum length of random walks considered. $\boldsymbol{R}_{i,j}$ is updated by an elementwise MLP $\Phi : \mathbb{R}^C \to \mathbb{R}^d$ to get the relative positional encoding $\boldsymbol{e}_{i,j}$, which encodes the structural relationship between node $i$ and node $j$.

We adopt attention design of the graph transformer introduced by Ma et al. (2023). Note that the graph transformer adapts self attention on $\boldsymbol{h}_i := \mathsf{H}_i[l, k, :] \in \mathbb{R}^d$ ($l \in [0, L-1]; k \in [0, K-1]$) of each index $[l, k]$ independently. Given $\boldsymbol{h}_i^{(0)} = \boldsymbol{h}_i$, $\boldsymbol{e}_{i,j}^{(0)} = \boldsymbol{e}_{i,j}$, the $t$-th layer of graph transformer ($t < T$) can be formulated as:

$$\hat{\boldsymbol{e}}_{i,j}^{(t)} = \sigma(\rho((\boldsymbol{W}_{\mathrm{Q}}\boldsymbol{h}_i^{(t)} + \boldsymbol{W}_{\mathrm{K}}\boldsymbol{h}_j^{(t)}) \odot \boldsymbol{W}_{\mathrm{Ew}}\boldsymbol{e}_{i,j}^{(t)}) + \boldsymbol{W}_{\mathrm{Eb}}\boldsymbol{e}_{i,j}^{(t)}) \in \mathbb{R}^d \qquad (7a)$$

$$\alpha_{ij} = \mathrm{Softmax}_{j \in \mathbb{V}}(\boldsymbol{W}_{\mathrm{A}}\hat{\boldsymbol{e}}_{i,j}^{(t)}) \in \mathbb{R} \qquad (7b)$$

$$\boldsymbol{h}_i^{(t+1)} = \sum_{j \in \mathbb{V}} \alpha_{ij} \cdot \boldsymbol{W}_{\mathrm{V}}\boldsymbol{h}_j^{(t)} \in \mathbb{R}^d \qquad (7c)$$

where $\boldsymbol{W}_{\mathrm{Q}}, \boldsymbol{W}_{\mathrm{K}}, \boldsymbol{W}_{\mathrm{Ew}}, \boldsymbol{W}_{\mathrm{Eb}}, \boldsymbol{W}_{\mathrm{V}} \in \mathbb{R}^{d \times d}$ and $\boldsymbol{W}_{\mathrm{A}} \in \mathbb{R}^{1 \times d}$ are learnable weight matrices; $\odot$ indicates elementwise multiplication; and $\rho(\boldsymbol{x}) := (\mathrm{ReLU}(\boldsymbol{x}))^{1/2} - (\mathrm{ReLU}(-\boldsymbol{x}))^{1/2}$. We also include feed-forward module, residual connection and normalization in our implementation, but they are omitted here for simplicity, which are detailed shown in Appendix A.2. The representation $\mathsf{H}_i$ of node $i$ is derived by gathering $\boldsymbol{h}_i^{(T)}$ of each index $[l, k]$.

For node-level graph reasoning tasks, the `Input` graph representation $\mathsf{G} = \mathsf{H}_i$, where node $i$ is to be inferred. For graph-level graph reasoning tasks, the `Input` graph representation $\mathsf{G} = \sum_{i \in \mathbb{V}} \mathsf{H}_i / |\mathbb{V}|$ by mean-pooling on the graph.

## 3.4 GRAPH-ENHANCED PREFIX TUNING FOR LLMS

To produce a `Response` in human language for a graph reasoning task, LLMs utilize graph-enhanced tunable prefix derived from the graph representation $\mathsf{G}$ during the tuning process. Specifi-

cally, the graph-enhanced prefix **P** is obtained by applying a linear projection to the graph representation **G** as illustrated in Eq. (8), where $\boldsymbol{W}_{\mathrm{U}} \in \mathbb{R}^{d \times d^{\mathrm{M}}}$ is a matrix converting the dimension.

$$\mathbf{P} = \mathbf{G} \boldsymbol{W}_{\mathrm{U}} + \mathbf{B} \tag{8}$$

Then $\mathbf{P} \in \mathbb{R}^{L \times K \times d^{\mathrm{M}}}$ is prepended to each attention layer of the LLM as shown in Eqs. (3) and (4).

**Connection to Prefix Tuning**    It's worth noting that when $W_{\mathrm{U}}$ is a zero matrix, GraphLLM degenerates into vanilla prefix tuning as **P** = **G**0 + **B**. From this perspective, GraphLLM is an enhancement of prefix tuning. In GraphLLM, the LLM synergizes with the powerful graph transformer to incorporate additional context information crucial to graph reasoning into the prefix. Consequently, the LLM can produce appropriate response for the graph reasoning task by interpreting the contexts encapsulated within the graph-enhanced prefix.

## 4    EXPERIMENT

In this section, we aim to empirically substantiate three central hypotheses posited in this study.

- **Q1**: Does GraphLLM effectively enhance the graph reasoning ability of the LLM?
- **Q2**: Can GraphLLM address the issue of lengthy context caused by *Graph2Text* strategy?
- **Q3**: How does GraphLLM perform in terms of computational efficiency?

### 4.1    EXPERIMENTAL SETTINGS

**Graph Reasoning Tasks**    We follow the design of the graph reasoning tasks in Wang et al. (2023a), which proposes a series of graph reasoning tasks with varying complexity on randomly generated graphs. Note that in Wang et al. (2023a), the nodes are identified and described by a single number index simply. This over-simplification potentially hinders a comprehensive evaluation of the model's capabilities in node understanding. Consequently, we develop four graph reasoning tasks where each node has a textual entity description of around 50 tokens. These tasks can simultaneously test the abilities of node understanding and structure understanding, which are both crucial for graph reasoning tasks. We present the illustration of the graph tasks in Figure 3, and the dataset statistics are provided in Table 1.

Table 1: Statistics of the graph reasoning task datasets.

|  | Substructure Counting | Maximum Triplet Sum | Shortest Path | Bipartite Graph Matching |
|---|---|---|---|---|
| Avg. $|\mathbb{V}|$ / Avg. $|\mathcal{E}|$ | 15 / 22.3 | 15 / 26.6 | 20 / 32.4 | 20 / 14.0 |
| No. of Tokens in Node Desc. | 52-59 | 39-82 | 48-58 | 34-61 |

- **Task 1: Substructure Counting** Let $\mathcal{G} = \{\mathcal{V}, \mathcal{E}\}$ be a molecular graph, where each atom in $\mathcal{V}$ has a text description $\boldsymbol{d}_i$ that includes the element type of the atom. LLMs are tasked with counting the number of specific substructure, e.g. carbon-carbon-oxygen triangle.

- **Task 2: Maximum Triplet Sum** Let $\mathcal{G} = \{\mathcal{V}, \mathcal{E}\}$ be a friendship graph, where each person in $\mathcal{V}$ has a text description $\boldsymbol{d}_i$ that includes the age of the person. In this task, LLMs are instructed to identify the maximum cumulative age among all possible triplets formed by selecting a specific individual, their direct friends, and the friends of those friends.

- **Task 3: Shortest Path** Let $\mathcal{G} = \{\mathcal{V}, \mathcal{E}\}$ be a graph that represents interconnected wormholes. Each wormhole in $\mathcal{V}$ requires a different amount of dark matter for activation, which is included in the text description $\boldsymbol{d}_i$ of each node. Activating a wormhole enables spatial jumps to any connected wormhole. LLMs are required to compute the path from the starting wormhole to the destination wormhole that requires the least amount of dark matter.

- **Task 4: Bipartite Graph Matching** Let $\mathcal{G} = \{\mathcal{V}, \mathcal{E}\}$ be a graph that depicts the application relationship between applicants and jobs. An edge in $\mathcal{E}$ represents an applicant applying for a specific job. Each job can only accept one applicant and a job applicant can be appointed for only one job. The text description $\boldsymbol{d}_i$ of each node provides information about either the job or the

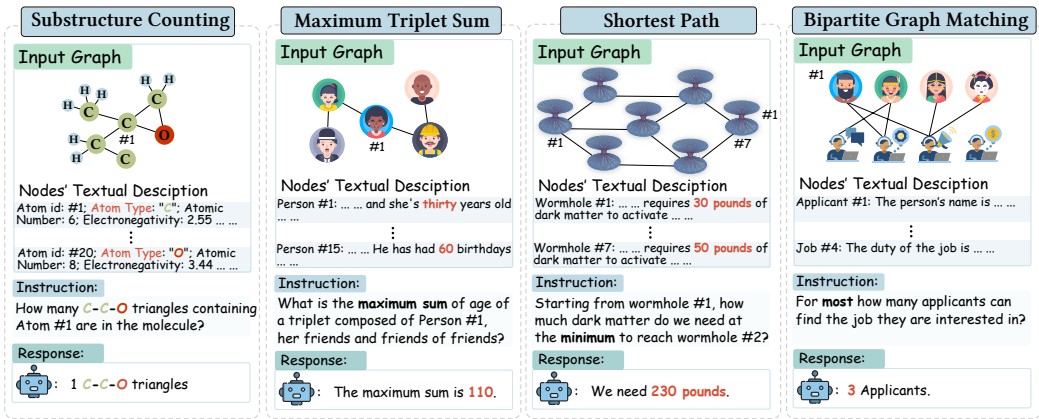

Figure 3: Illustration of the graph reasoning tasks. Each `Input` graph consists of a number of nodes characterized by textual node descriptions and the graph structure between the nodes.

applicant. LLMs are required to compute the maximum possible number of applicants who can find the jobs they are interested in.

Each task consists of 2,000/2,000/6,000 graph instance for training/validation/test. The textual descriptions of the nodes are generated by `gpt-3.5-turbo` according to specific instructions and manually verified. The graph descriptions of these tasks using *Graph2Text* strategy are presented in the Appendix D.

**Baselines** We compare GraphLLM with two categories of approaches: prompting and fine-tuning. The prompting approaches encompass the following strategies: zero-shot prompting, few-shot in-context learning (Brown et al., 2020) and few-shot chain-of-thought (CoT) prompting (Wei et al., 2022). The fine-tuning approaches include widely adopted prefix tuning (Li & Liang, 2021) and LoRA (Hu et al., 2022). Due to context length limit, all tasks are confined to one shot for few-shot methods. For LoRA, we apply low rank adaption only on attention module (attn) and on both attention module and feed-forward networks (attn+ffn) (Zhang et al., 2023). For all the baselines, we follow Wang et al. (2023a); Guo et al. (2023) to design prompts which describe the `Input` graph in natural language (*Graph2Text*). To analyze the performance gap that may emerge from utilizing different graph description languages, we utilized two prevalent methods to describe the graph structure: adjacency list and edge list.

**Imeplementations** We use **LLaMA 2** 7B/13B (Touvron et al., 2023) as our LLM backbone. For all tested methods, we set the temperature $\tau$ to 0 to ensure that the LLM's response is deterministic. We adopt Exact Match Accuracy as metrics for the four graph reasoning tasks. All experiments are conducted on $4 \times 80G$ A100 GPUs. Complete experiment setups such as hyperparameters, batch size, optimizer, learning rates are in Appendix B.

Table 2: Performance on Graph Reasoning Tasks. Shown is the mean ± s.d. of 3 runs with different random seeds. Highlighted are the top and second-best.

| Input Format | Method | LLaMA2-7B | | | | LLaMA2-13B | | | |
|---|---|---|---|---|---|---|---|---|---|
| | | Substructure Counting | Maximum Triplet Sum | Shortest Path | Bipartite Graph Matching | Substructure Counting | Maximum Triplet Sum | Shortest Path | Bipartite Graph Matching |
| Adjacency List | Zero-shot | 0.2260 | 0.1110 | 0.0000 | 0.3630 | 0.0145 | 0.0925 | 0.0010 | 0.1180 |
| | Few-shot | 0.2735 | 0.1445 | 0.0575 | 0.3280 | 0.2780 | 0.1430 | 0.0520 | 0.2675 |
| | Few-shot CoT | 0.2177 | 0.0585 | 0.1089 | 0.2399 | 0.2150 | 0.0544 | 0.1552 | 0.1048 |
| | LoRA(attn) | $0.5012_{\pm.0054}$ | $0.4427_{\pm.0031}$ | $0.2119_{\pm.0004}$ | $0.7383_{\pm.1078}$ | $0.4926_{\pm.0068}$ | $0.4080_{\pm.0009}$ | $0.1251_{\pm.0019}$ | $0.7792_{\pm.0353}$ |
| | LoRA(attn+ffn) | $0.5400_{\pm.0363}$ | $0.4723_{\pm.0115}$ | $0.1652_{\pm.0420}$ | $0.6941_{\pm.0691}$ | $0.4948_{\pm.0035}$ | $0.4274_{\pm.0459}$ | $0.1181_{\pm.0051}$ | $0.8010_{\pm.0490}$ |
| | Prefix Tuning | $0.5003_{\pm.0134}$ | $0.3887_{\pm.0346}$ | $0.2173_{\pm.0078}$ | $0.5534_{\pm.0739}$ | $0.4610_{\pm.0444}$ | $0.3377_{\pm.0038}$ | $0.1608_{\pm.0376}$ | $0.4640_{\pm.0314}$ |
| Edge List (Random Order) | Zero-shot | 0.2460 | 0.1260 | 0.0000 | 0.4325 | 0.0805 | 0.1265 | 0.0010 | 0.0055 |
| | Few-shot | 0.2610 | 0.1420 | 0.0111 | 0.3687 | 0.2655 | 0.1423 | 0.1110 | 0.3230 |
| | Few-shot CoT | 0.2127 | 0.0565 | 0.1069 | 0.1411 | 0.2320 | 0.0767 | 0.1351 | 0.0464 |
| | LoRA(attn) | $0.5035_{\pm.0007}$ | $0.4224_{\pm.0040}$ | $0.2011_{\pm.0074}$ | $0.6457_{\pm.0243}$ | $0.4920_{\pm.0172}$ | $0.4143_{\pm.0059}$ | $0.1240_{\pm.0008}$ | $0.6319_{\pm.0199}$ |
| | LoRA(attn+ffn) | $0.5101_{\pm.0051}$ | $0.4552_{\pm.0319}$ | $0.2011_{\pm.0046}$ | $0.5446_{\pm.0364}$ | $0.4904_{\pm.0051}$ | $0.4489_{\pm.0157}$ | $0.1958_{\pm.0180}$ | $0.6126_{\pm.0338}$ |
| | Prefix Tuning | $0.3925_{\pm.0612}$ | $0.3780_{\pm.0131}$ | $0.1656_{\pm.0273}$ | $0.4599_{\pm.0187}$ | $0.3319_{\pm.1148}$ | $0.3525_{\pm.0048}$ | $0.1246_{\pm.0014}$ | $0.5228_{\pm.0575}$ |
| | GraphLLM | $0.9990_{\pm.0007}$ | $0.9577_{\pm.0058}$ | $0.9726_{\pm.0011}$ | $0.9981_{\pm.0015}$ | $0.9890_{\pm.0021}$ | $0.9392_{\pm.0064}$ | $0.9619_{\pm.0038}$ | $0.9934_{\pm.0064}$ |

## 4.2 PERFORMANCE ON GRAPH REASONING TASKS (Q1)

Table 2 delineates the performance differentials between GraphLLM and *Graph2Text*-based methodologies across the four graph reasoning tasks. From this comparative analysis, we can infer several key insights: (1). The zero-shot, few-shot, and chain-of-thought *Graph2Text*-based prompting methods deliver subpar performance, indicating the limitations of LLMs in generalizing to graph reasoning tasks without additional fine-tuning. (2). Even with fine-tuning on graph reasoning tasks, *Graph2Text*-based methodology significantly lag behind the performance achieved by GraphLLM. This discrepancy suggests that the *Graph2Text*-based approaches can constitute a significant obstacle preventing LLMs from adapting to graph reasoning tasks. (3). The choice between the two primary graph description languages (adjacency/edge list) doesn't lead to a consistent enhancement in the performance of *Graph2Text*-based methods. This finding confirms that the impediments introduced by the *Graph2Text* methodology aren't tied to a specific graph description language. (4). On average, GraphLLM achieves an Exact Match Accuracy of **98.19**% over the four tasks, in contrast to the top-performing *Graph2Text*-based method, which manages only 47.35%. This difference underscores the effectiveness of our approach in facilitating LLMs in graph reasoning tasks.

**Evaluation on Stronger LLMs** We also evaluate the *Graph2Text* strategy on more powerful `gpt-3.5-turbo` and `gpt-4`, illustrated on Table 3. The results indicate that even the advanced `gpt-4` falls short in basic graph reasoning tasks, limiting its application in more complex scenarios such as drug design. GraphLLM provides a lightweight fine-tuning method that enables the LLM to synergize with graph reasoning modules. Notably, GraphLLM with LLaMA 2 7B as the backbone LLM shows relative improvements of 2.61%, 99.8%, 12.22%, and 15.16% compared to `gpt-4` few-shot CoT on the four fundamental graph reasoning tasks, respectively.

Table 3: Performance of `gpt-3.5-turbo` and `gpt-4` with *Graph2Text* strategy (converting input graph into adjacency list described in natural language), evaluated on 30 random samples due to the money cost.

| LLM | Method | Substructure Counting | Maximum Triplet Sum | Shortest Path | Bipartite Graph Matching |
|---|---|---|---|---|---|
| `gpt-3.5-turbo` | Zero-shot | 0.2667 | 0.5667 | 0.2000 | 0.1000 |
| | Few-shot | 0.3000 | 0.3000 | 0.2667 | 0.0667 |
| | Few-shot CoT | 0.3667 | 0.7000 | 0.7333 | 0.2667 |
| `gpt-4` | Zero-shot | 0.6000 | 0.7333 | 0.6667 | 0.3333 |
| | Few-shot | 0.5000 | 0.8667 | 0.5667 | 0.5000 |
| | Few-shot CoT | 0.5000 | 0.9333 | 0.8667 | 0.8667 |
| `LLaMA 2-7B` | GraphLLM | 0.9990 | 0.9577 | 0.9726 | 0.9981 |

## 4.3 COMPARATIVE ANALYSIS ON CONTEXT REDUCTION (Q2)

Table 4 demonstrates the LLM context length for graph reasoning tasks utilizing *Graph2Text*-based methods and GraphLLM, respectively. Notably, GraphLLM reduces the context length by a substantial **96.45**% across the four graph reasoning tasks averagely. This substantial reduction is achieved as GraphLLM encodes both node descriptions and structural information into a fixed-length prefix (5 additional prefix tokens in our GraphLLM's implementation). In contrast, *Graph2Text*-based methods describe the graph in natural language, including both node descriptions and graph structure. This approach inherently results in an extended context, potentially hampering the efficiency and effectiveness of LLMs on graph reasoning.

Figure 4 illustrates the performance of *Graph2Text*-based methods and GraphLLM on the substructure counting task when the size of graph increases. More concretely, the average node number of the graph instances in the substructure counting dataset is incrementally increased from 15 to 45, with a step size of 10. We compare GraphLLM with *Graph2Text*-based prefix tuning and LoRA, ignoring other less effective baseline methods. We additionally compare GraphLLM with *Graph2Text*-based few-shot CoT on `gpt-3.5-turbo-16k`, because the context limit of `gpt-3.5-turbo`/`gpt-4` is exceeded when the node number reaches 25. We observe that with the increase in graph size, the context size of the *Graph2Text*-based method also expands, leading to a corresponding decline in performance. It is noteworthy that as the graph size increases to 45 nodes, *Graph2Text*-based methods with LLaMA 2 as backbone exceeds the context length limit (4096 tokens), and the performance of `gpt-3.5-turbo-16k` also dropped to 0. In comparison, GraphLLM still retains an accuracy of 0.9645. This stability highlights the robustness of GraphLLM, contrasting with the declining performance and efficiency observed in *Graph2Text*-based methods as graph size expands.

Table 4: Context length of different methods on graph reasoning tasks, measured by average token number processed by the LLaMA 2 tokenizer. A/B shown is the context length of *Graph2Text*-based methods with adjacency list/edge list as graph description language.

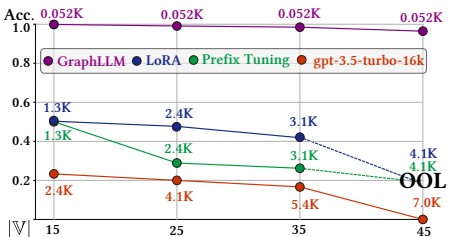

| Method | Avg. Context Length | | | |
|---|---|---|---|---|
| | Substructure Counting | Maximum Triplet Sum | Shortest Path | Bipartite Graph Matching |
| Zero-shot | 1.3K / 1.3K | 1.4K / 1.4K | 1.8K / 1.7K | 1.2K / 1.2K |
| Few-shot | 2.6K / 2.5K | 2.8K / 2.8K | 3.1K / 2.9K | 2.4K / 2.7K |
| Few-shot CoT | 2.8K / 2.7K | 3.0K / 2.9K | 3.3K / 3.1K | 2.5K / 2.8K |
| LoRA | 1.3K / 1.3K | 1.4K / 1.4K | 1.8K / 1.7K | 1.2K / 1.2K |
| Prefix Tuning | 1.3K / 1.3K | 1.4K / 1.4K | 1.8K / 1.7K | 1.2K / 1.2K |
| GraphLLM | 0.040K (↓**96.92%**) | 0.052K (↓**96.29%**) | 0.048K (↓**97.18%**) | 0.055K (↓**95.42%**) |

Figure 4: Performance on substructure counting tasks when increasing the node number $|\mathbb{V}|$ of graph instances. A(B) represents context length and the corresponding performance. "OOL" denotes exceeding context length limit.

## 4.4 ANALYSIS ON COMPUTATIONAL EFFICIENCY (Q3)

**Inference Acceleration** Figure 5 illustrates the comparison of inference times on substructure counting task between GraphLLM and *Graph2Text*-based methods. Notably, GraphLLM achieves a speedup of **3.42** times compared to the best-performing *Graph2Text*-based method. The complete results of the inference time for other tasks are provided in the Appendix. The experimental results indicate that the inference acceleration achieved by GraphLLM, due to the context reduction for graph reasoning tasks, considerably surpasses the additional time overhead introduced by the graph learning module.

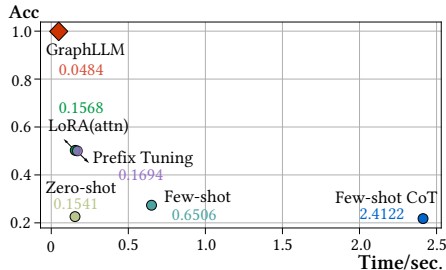

Figure 5: Avg. inference time on the substructure counting task on LLaMA 2 7B .

## 5 RELATED WORK

LLMs exhibit the ability to understand diverse types of information and craft contextually relevant text responses, including but not limited to images (Wang et al., 2023b), audio (Huang et al., 2023), and point clouds (Xu et al., 2023). Endeavors to empower LLMs with the ability to understand graph data have been ongoing. Generally, these efforts can be categorized into two main categories. The first category includes models that employ a large language model to interface with individual graph models or APIs (Zhang, 2023; Wei et al., 2023). Nevertheless, these interactive systems still encounter limitations in accessing the internal graph reasoning process, which hinder their ability to seamlessly integrate graph learning and large language models. The second category includes models that employ an end-to-end training strategy. Notably, Wang et al. (2023a) make an attempt to fine-tune an opt-2.5B model on a *Graph2Text* corpus of basic graph reasoning tasks. However, their efforts fail to elicit graph reasoning ability of LLMs. The task of enhancing the graph reasoning ability of LLMs in an end-to-end manner remains unresolved. To our knowledge, our work stands out as a pioneering effort in successfully integrating the graph learning model with LLMs, demonstrably enhancing graph reasoning ability. GraphLLM takes a unified, end-to-end approach to integrate graph learning models and LLMs, enhancing the overall efficiency by synergizing the strengths of both within a single, cohesive system.

## 6 DISCUSSION

We introduce GraphLLM, an integrated end-to-end approach that synergizes LLMs with graph learning models to enhance the graph reasoning capabilities of LLMs. We hope our work can provide insights and guidance for future research in the domain of enabling LLMs to comprehend graph data and tackle advanced graph-related tasks.

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

# A  DETAILED FORMULATION OF GRAPHLLM

## A.1  DETAILS OF TEXTUAL TRANSFORMER ENCODER-DECODER

Details of the textual transformer encoder-decoder architecture are shown in Figure 6. In each layer of the transformer encoder, the sequential node textual features pass through the multi-head self-attention module without masking, allowing for a contextual understanding of the text sequence. The resulting encoded sequence, $c_i$, engages in the cross-attention with fixed-size query embeddings in the transformer decoder. This process enables query embeddings to extract essential information from it. Finally, the output of the transformer decoder, $\mathbf{H}_i$, contains specific information from the encoded sequence $c_i$, serving as the node representation.

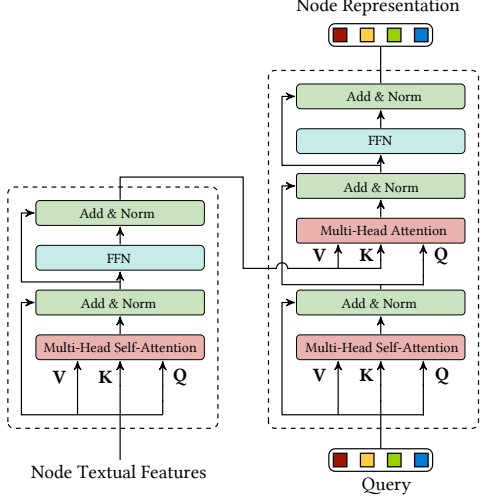

Figure 6: Architecture of the textual transformer encoder-decoder in GraphLLM.

## A.2  COMPLETE FORMULATION OF GRAPH TRANSFORMER

A complete graph transformer layer comprises a multi-head attention module, a feed-forward network, along with the residual connection and layer normalization associated with each of these components. For the $t$-th layer in the graph transformer, the attention computation, excluding the multi-head part, is as follows:

$$\hat{e}_{i,j}^{(t)} = \sigma(\rho((\boldsymbol{W}_{\mathrm{Q}}\boldsymbol{h}_i^{(t)} + \boldsymbol{W}_{\mathrm{K}}\boldsymbol{h}_j^{(t)}) \odot \boldsymbol{W}_{\mathrm{Ew}}\boldsymbol{e}_{i,j}^{(t)}) + \boldsymbol{W}_{\mathrm{Eb}}\boldsymbol{e}_{i,j}^{(t)}) \in \mathbb{R}^d \tag{9a}$$

$$\alpha_{ij} = \mathrm{Softmax}_{j \in \mathbb{V}_i}(\boldsymbol{W}_{\mathrm{A}}\hat{e}_{i,j}^{(t)}) \in \mathbb{R} \tag{9b}$$

$$\hat{\boldsymbol{h}}_i^{(t)} = \sum_{j \in \mathbb{V}_i} \alpha_{ij} \cdot \boldsymbol{W}_{\mathrm{V}}\boldsymbol{h}_j^{(t)} \in \mathbb{R}^d \tag{9c}$$

where $\boldsymbol{W}_{\mathrm{Q}}, \boldsymbol{W}_{\mathrm{K}}, \boldsymbol{W}_{\mathrm{Ew}}, \boldsymbol{W}_{\mathrm{Eb}}, \boldsymbol{W}_{\mathrm{V}} \in \mathbb{R}^{d \times d}$ and $\boldsymbol{W}_{\mathrm{A}} \in \mathbb{R}^{1 \times d}$ are learnable weight matrices; $\sigma$ is a non-linear activation (ReLU by default); $\rho(\boldsymbol{x}) := (\mathrm{ReLU}(\boldsymbol{x}))^{1/2} - (\mathrm{ReLU}(-\boldsymbol{x}))^{1/2}$; $\odot$ indicates elementwise multiplication.

The different attention heads are combined as a whole, and this combination is then subject to a residual connection and passed through layer normalization to obtain the output of the multi-head attention module.

$$\boldsymbol{h}_i^{(t),\mathrm{attn}} = \mathtt{LayerNorm}(\mathrm{Concat}(\{\hat{\boldsymbol{h}}_{i,h}^{(t)}\}_{h=1}^{N_h})\boldsymbol{W}_{\mathrm{O}} + \boldsymbol{h}_i^{(t)}) \tag{10a}$$

$$\boldsymbol{e}_{i,j}^{(t),\mathrm{attn}} = \mathtt{LayerNorm}(\mathrm{Concat}(\{\hat{\boldsymbol{e}}_{i,j,h}^{(t)}\}_{h=1}^{N_h})\boldsymbol{W}_{\mathrm{Eo}} + \boldsymbol{e}_{i,j}^{(t)}) \tag{10b}$$

where $\boldsymbol{W}_{\mathrm{O}}, \boldsymbol{W}_{\mathrm{Eo}} \in \mathbb{R}^{d \times d}$ are learnable weight matrices, $N_h$ denotes the number of attention heads and $\boldsymbol{h}_i^{(t),\mathrm{attn}}, \boldsymbol{e}_{i,j}^{(t),\mathrm{attn}}$ are the normalized outputs of the attention module.

The feed-forward network, the corresponding residual connection and layer normalization can be formulated as:

$$h_i^{(t+1)} = \texttt{LayerNorm}(\texttt{Feedforward}(h_i^{(t),\text{attn}}) + h_i^{(t),\text{attn}}) \tag{11a}$$

$$e_{i,j}^{(t+1)} = \texttt{LayerNorm}(\texttt{Feedforward}(e_{i,j}^{(t),\text{attn}}) + e_{i,j}^{(t),\text{attn}}) \tag{11b}$$

where $h_i^{(t+1)}, e_{i,j}^{(t+1)}$ are the outputs of the entire $t$-th graph transformer layer.

## B  SETUP

We provide the hyperparameters of our method for different graph tasks in Table 5. For the baseline methods that require fine-tuning of LLM, we ensure fair comparison by training them for the same number of epochs as GraphLLM. Additionally, we conducted a search for some important hyperparameters. Specifically, we search the rank parameter of the LoRA from a set $\{4, 8, \mathbf{16}\}$ and the number of prefix tokens in prefix tuning from a set $\{5, \mathbf{10}, 20\}$.

Table 5: Hyperparameters of GraphLLM for the four datasets.

| Hyperparameter | Substructure Counting | Maximum Triplet Sum | Shortest Path | Bipartite Graph Matching |
|---|---|---|---|---|
| Textual Encoder | 4 | 4 | 4 | 4 |
| Textual Decoder | 4 | 4 | 4 | 4 |
| Graph Transformer | 4 | 4 | 4 | 4 |
| Hidden dim | 768 | 768 | 768 | 768 |
| Heads | 6 | 6 | 6 | 6 |
| Dropout | 0 | 0 | 0 | 0 |
| Graph pooling | - | - | - | mean |
| Prefix | 5 | 5 | 5 | 5 |
| PE dim | 8 | 8 | 8 | 8 |
| Batch size | 32 | 32 | 32 | 32 |
| Learning Rate | $5e-5$ | $5e-5$ | $5e-5$ | $5e-5$ |
| Epochs | 15 | 20 | 20 | 15 |
| Warmup epochs | 1 | 1 | 1 | 1 |
| Weight decay | $1e-1$ | $1e-1$ | $1e-1$ | $1e-1$ |
| Tunable parameters | 0.0933B | 0.0933B | 0.0933B | 0.0933B |

## C  SUPPLEMENTAL EXPERIMENT RESULTS

### C.1  INFERENCE TIME

In Table 6, we provide the inference time of different methods on LLaMA 2 7B and 13B. The results are calculated by taking the average inference time of all instances in the test set. From the results, we can observe that due to the reduction in context for graph reasoning tasks, GraphLLM exhibits a significant advantage in terms of inference time compared to *Graph2Text*-based methods. Furthermore, this advantage becomes more pronounced as the LLM's scale increases.

### C.2  ABLATION STUDY ON GRAPH TRANSFORMER

We experiment with different design choices on the structure understanding module. Specifically, we replace the aggregation mechanism via attention in graph transformer with other commonly used graph learning layer. Here we adopt GIN (Xu et al., 2019) and GAT (Veličković et al., 2018), while keeping the other modules unchanged in each case. Table 7 shows the experimental results on the four graph reasoning tasks. GIN variant and GAT variant only achieve average accuracies of 24.2% and 17.3%, respectively. The significant disparity in accuracy between GIN variant, GAT variant,

Table 6: Inference time on the four graph reasoning tasks.

| Input Format | Method | LLaMA2-7B | | | | LLaMA2-13B | | | |
|---|---|---|---|---|---|---|---|---|---|
| | | Maximum Path Sum | Substructure Counting | Shortest Path | Bipartite Graph Matching | Maximum Path Sum | Substructure Counting | Shortest Path | Bipartite Graph Matching |
| Adjacency List | Zero-shot | 0.1673 | 0.1541 | 0.2649 | 0.1385 | 0.2878 | 0.2670 | 0.4517 | 0.2402 |
| | Few-shot | 0.4269 | 0.6506 | 0.5168 | 0.6116 | 0.7479 | 1.1150 | 0.8848 | 1.0604 |
| | CoT | 4.9367 | 2.4122 | 5.5155 | 4.2542 | 8.2850 | 4.1148 | 9.2961 | 7.2886 |
| | LoRA(attn) | 0.1568 | 0.1568 | 0.2804 | 0.1420 | 0.2926 | 0.2698 | 0.4601 | 0.2455 |
| | LoRA(attn+ffn) | 0.1818 | 0.1654 | 0.2842 | 0.1503 | 0.3071 | 0.2842 | 0.4813 | 0.2586 |
| | Prefix Tuning | 0.1846 | 0.1694 | 0.2947 | 0.1519 | 0.3161 | 0.2910 | 0.4963 | 0.2603 |
| Edge List (Random Order) | Zero-shot | 0.1642 | 0.1447 | 0.2521 | 0.1486 | 0.2869 | 0.2508 | 0.4355 | 0.2573 |
| | Few-shot | 0.4216 | 0.6259 | 0.4938 | 0.6560 | 0.7350 | 1.0678 | 0.8457 | 1.1473 |
| | CoT | 4.8385 | 2.3190 | 5.3227 | 4.5724 | 8.1369 | 3.9703 | 9.0008 | 7.8130 |
| | LoRA(attn) | 0.1678 | 0.1465 | 0.2569 | 0.1511 | 0.2923 | 0.2539 | 0.4431 | 0.2622 |
| | LoRA(attn+ffn) | 0.1783 | 0.1556 | 0.2714 | 0.1597 | 0.3054 | 0.2670 | 0.4636 | 0.2758 |
| | Prefix Tuning | 0.1777 | 0.1623 | 0.2757 | 0.1632 | 0.3080 | 0.2821 | 0.4724 | 0.2825 |
| | GraphLLM | 0.0449 | 0.0484 | 0.0734 | 0.0523 | 0.0583 | 0.0616 | 0.0937 | 0.0665 |

and GraphLLM indicates that the practice of decoupling node information and structural information plays an essential role in improving GraphLLM's structure understanding ability, subsequently enhancing the graph reasoning capability.

Table 7: Ablation study on graph transformer.

| Ablation | Maximum Triplet Sum | Substructure Counting | Shortest Path | Bipartite Graph Matching |
|---|---|---|---|---|
| GT → GINConv | $0.2237_{\pm.0060}$ | $0.3878_{\pm.0650}$ | $0.2122_{\pm.0053}$ | $0.1427_{\pm.0019}$ |
| GT → GATConv | $0.1819_{\pm.0053}$ | $0.2598_{\pm.0102}$ | $0.1443_{\pm.0017}$ | $0.1052_{\pm.0015}$ |
| GraphLLM | $0.9577_{\pm.0058}$ | $0.9990_{\pm.0007}$ | $0.9726_{\pm.0011}$ | $0.9981_{\pm.0015}$ |

# D   EXAMPLES OF GRAPH REASONING TASKS

## Substructure Counting

---

**Input:**

Here are the descriptions of 15 atoms in a molecule.

Atom 1: The Carbon atom has an atomic number of 6, denoted as "C". Carbon has an electronegativity value of approximately 3.25. The covalent radius of a Carbon atom is about ...

Atom 2: The Carbon atom has an atomic number of 6, denoted as "C". Carbon has an electronegativity value of approximately 2.16. The covalent radius of a Carbon atom is about ...

Atom 3: The Oxygen atom has an atomic number of 8, denoted as "O". Oxygen has an electronegativity value of approximately 3.52. The covalent radius of a Oxygen atom is about ...

...

Atom 15: The Nitrogen atom has an atomic number of 7, denoted as "N". Nitrogen has an electronegativity value of approximately 2.95. The covalent radius of a Nitrogen atom is ...

These atoms are connected as the following undirected graph to form the molecule:

Atom 1 is connected with: Atom 2, Atom 3, Atom 4, Atom 5, Atom 6, Atom 7, Atom 8.

Atom 2 is connected with: Atom 1, Atom 3, Atom 5, Atom 8.

Atom 3 is connected with: Atom 1, Atom 2.

...

Atom 15 is connected with: Atom 8, Atom 14.

How many carbon-carbon-oxygen triangles containing Atom 1 are in the molecule?

**Response on** `gpt-4` **Failure Case :**

A carbon-carbon-oxygen triangle would consist of two carbon atoms and one oxygen atom. Since Atom 1 is a carbon atom and must be included in the triangle, we are looking for triangles where the other two atoms are one carbon and one oxygen.

Looking at the connections of Atom 1, it is connected to the following carbon atoms: Atom 2, Atom 4, Atom 6, Atom 7. It is also connected to the following oxygen atoms: Atom 3, Atom 5, Atom 8.

Now we need to see if any of these carbon atoms are connected to any of these oxygen atoms.

- Atom 2 (carbon) is connected to Atom 3 (oxygen) and Atom 8 (oxygen).

- Atom 4 (carbon) is not connected to any of the oxygen atoms connected to Atom 1.

- Atom 6 (carbon) is not connected to any of the oxygen atoms connected to Atom 1.

- Atom 7 (carbon) is not connected to any of the oxygen atoms connected to Atom 1.

So, there are **two** carbon-carbon-oxygen triangles that contain Atom 1: Atom 1 - Atom 2 - Atom 3 and Atom 1 - Atom 2 - Atom 8.

**Correct Answer:**

There are **3** C-C-O triangles containing Atom 1.

---

## Maximum Triplet Sum

**Input:**

Here are the descriptions of 15 people.

Person 1: She is Wilma Lyons, and she is a sixty-year-old. With her colorful hair and unconventional fashion sense, she stands out as a true original. Her unassuming nature and humility create an environment ...

Person 2: Meet Manuel Cornelius, who is 30 years of age. With her adventurous spirit and love for the outdoors, she's always up for exploring new places and experiences. She possesses an air of sophistication and grace, seen in her timeless fashion ...

...

Person 15: Her petite frame and delicate features give her a dainty and ethereal presence. Her name is Cornelia Brooks, and she's forty years old. Her contagious enthusiasm and energy inspire others to ...

The relationship between them can be described as the following undirected graph:

Person 1 is connected with: Person 14, Person 8.

Person 2 is connected with: Person 11, Person 14, Person 15.

Person 3 is connected with: Person 11, Person 5, Person 13, Person 6.

...

Person 15 is connected with: Person 2, Person 7, Person 9, Person 10, Person 11.

Question: What is the maximum sum of age of a triplet composed of Person 1, her friends and friends of friends?

**Response on** `gpt-4` **Failure Case :**

Person 1 is connected with Person 14 and Person 8.

Person 14 is connected with Person 1, Person 2, Person 7, Person 12, Person 13. Among these, Person 2, Person 7, and Person 12 are not connected with Person 1, so they can be the third person on the path. Their ages are 50, 30, and 60 respectively.

Person 8 is connected with Person 1, Person 11, Person 6. Among these, Person 11 and Person 6 are not connected with Person 1, so they can be the third person on the path. Their ages are 40 and 40 respectively.

So the maximum sum of the 3 people's age on the path is 60 (Person 1) + 60 (Person 14) + 60 (Person 12) = **180**.

**Correct Answer:**

The maximum sum is **160**.

## Shortest Path

**Input:**

Here are the descriptions of 20 wormholes.

It is wormhole 1, and it is located in galaxy ARP 188. This wormhole is about 5463 light-years away from Earth and requires 20 pounds of dark matter to activate.

It is wormhole 2, and it is located in galaxy Horsehead Nebula. This wormhole is about 7606 light-years away from Earth and requires 20 pounds of dark matter to activate.

It is wormhole 3, and it is located in galaxy Large Magellanic Cloud. This wormhole is about 4214 light-years away from Earth and requires 40 pounds of dark matter to activate.

It is wormhole 4, and it is located in galaxy Pelican Nebula. This wormhole is about 3920 light-years away from Earth and requires 40 pounds of dark matter to activate.

...

It is wormhole 20, and it is located in galaxy Needle Galaxy. This wormhole is about 844 light-years away from Earth and requires 30 pounds of dark matter to activate.

These wormholes are connected as the following undirected graph:

Wormhole 1 is connected with: Wormhole 8, Wormhole 9, Wormhole 12, Wormhole 19.

Wormhole 2 is connected with: Wormhole 3, Wormhole 13, Wormhole 17, Wormhole 20.

Wormhole 3 is connected with: Wormhole 2, Wormhole 7, Wormhole 8, Wormhole 13.

...

Wormhole 20 is connected with: Wormhole 2, Wormhole 9, Wormhole 13.

Question: Starting from wormhole 1, How much dark matter we'll need at the minimum to reach Wormhole 2?

**Response on** `gpt-4` **Failure Case :**

From the descriptions, we know that:

- Wormhole 1 requires 20 pounds of dark matter to activate.

- Wormhole 8 requires 20 pounds of dark matter to activate.

- Wormhole 3 requires 40 pounds of dark matter to activate.

- Wormhole 2 requires 20 pounds of dark matter to activate.

Adding these up, you need a total of 20 + 20 + 40 + 20 = **100** pounds of dark matter to reach wormhole 2 from wormhole 1.

**Correct Answer:**

We need **80** pounds.

## Bipartite Graph Matching

**Input:**

Here are the descriptions of 10 job applicants.

Applicant 1: She is Adam Lamarr, and she is 51 years old. She wants to find a job. She's an urban planner, designing sustainable cities, harmonizing architecture and environment for better living.

...

Applicant 10: He is Travis Wight, and he is 18 years old. He wants to find a job. He finds peace in practicing meditation and mindfulness, nurturing his well-being.

Here are the descriptions of 10 jobs.

Job 1: The average salary for a year is 47326.6 dollars, and it needs to work 38 hours every week. We want applicants with related experience.

...

Each applicant is interested in some of the jobs, and the relationship can be described as the following graph.

Applicant 1 is interested in: Job 9.

Applicant 2 is interested in: Job 6.

...

Applicant 10 is interested in: Job 6.

Each job can only accept one applicant and a job applicant can be appointed for only one job. For most how many applicants can find the job they are interested in?

**Response on `gpt-4` Failure Case :**

First, let's look at the jobs that only have one applicant interested in them. These are Job 9, Job 4, and Job 10. Applicant 1, Applicant 3, Applicant 8, and Applicant 7 can be appointed to these jobs respectively.

Next, let's look at the jobs that have multiple applicants interested in them. These are Job 1, Job 5, Job 6, and Job 7.

For Job 1, the only remaining interested applicant is Applicant 7, so she can be appointed to this job.

For Job 5, the only interested applicant is Applicant 5, so he can be appointed to this job.

For Job 6, there are three interested applicants: Applicant 2, Applicant 5, and Applicant 9. Since Applicant 5 has already been appointed to Job 5, we can appoint Applicant 2 to Job 6.

For Job 7, the only interested applicant is Applicant 5, but he has already been appointed to Job 5. Therefore, Job 7 will remain vacant.

In conclusion, the maximum number of applicants that can find the job they are interested in is **6**: Applicant 1, Applicant 3, Applicant 8, Applicant 7, Applicant 5, and Applicant 2.

**Correct Answer:**

**5** Applicants.

