# OpenReview forum: "GraphLLM: Boosting Graph Reasoning Ability of Large Language Model"
_ICLR.cc/2024/Conference — Submitted to ICLR 2024_

### Official Review · Reviewer_DjB6 · 2023-10-30

**Soundness:** 2 fair
**Presentation:** 3 good
**Contribution:** 2 fair
**Rating:** 3
**Confidence:** 4

**Summary:**

In this paper, the authors propose GraphLLM to enhance the graph reasoning ability of LLMs. Specifically, GraphLLM is an end-to-end approach that integrates LLMs with graph Transformer modules. Compared to existing graph2text approaches, the proposed method can improve accuracy and reduce context length. Experimental results on four graph reasoning tasks, including substructure counting, maximum triplet sum, shortest path, and bipartite graph matching demonstrate the effectiveness of the proposed method.

**Strengths:**

1. Applying LLMs to graph-related problems is a recently emerging topic with rich potential.
2. The proposed method can combine graph machine learning (i.e., graph Transformer) and LLMs to better capture the graph structural information.
3. Experiments demonstrate the efficacy of the proposed method in improving accuracy and reducing context length.
4. The authors have released the codes to facilitate reproducibility.

**Weaknesses:**

1. I would strongly suggest applying the proposed method besides simple artificial graph tasks, where off-the-shelf algorithms already exist to perfectly solve them. Experiments on more real-world tasks such as node classification, link prediction, graph classification, etc., can better demonstrate the effectiveness of the proposed method. This is especially important for a technical paper where new approaches are introduced, as opposed to the previous observation-based papers such as NLGraph and GPT4Graph.
2. One major advantage of using LLMs compared to GNNs is explainability. Therefore, I would recommend conducting some analyses in this aspect. For example, if the proposed method cannot produce reasons for its prediction, it is likely that the proposed method captures some spurious correlations and does not truly “solve” the problem.
3. I also wonder whether the authors have considered directly comparing with graph Transformers or GNNs, to see whether LLMs have advanced the field.
4. Based on my understanding, one major advantage of the Graph2Text pipeline is being able to utilize close-sourced LLMs such as GPT-3.5 or GPT-4 without any fine-tuning step, since we only need to input data and related information (prompts, queries, etc.) to APIs. In comparison, as the proposed method needs to be trained end-to-end, it increases the computation cost and can only operate with open-sourced LLMs (in order to obtain gradients), which limits the applicability.
5. All things considered, the technical contribution of the paper is okay but not too novel, considering that all three components are heavily based on the existing literature, i.e., node understanding is a standard Transformer, structure understanding is essentially equivalent to Ma et al., 2023, and prefix tuning follows the common practice of LLMs. It would make the paper stronger if these components could be customized to further enhance the performance.
6. Minor: the authors claim one advantage of graph Transformers over GNNs is “its decoupling of node information and structural information”. Experimental evidence could be provided to support this claim.

**Questions:**

See Weaknesses above

---

> ### Author Response · Authors · 2023-11-19
> **Response to Reviewer DjB6 [1/3]**
>
> Dear reviewer DjB6,
>
> Thank you for your review and valuable comments. We have carefully considered your feedback and provide here our detailed responses to each point.
>
> > Q1  I would strongly suggest applying the proposed method besides simple artificial graph tasks, where off-the-shelf algorithms already exist to perfectly solve them. Experiments on more real-world tasks such as node classification, link prediction, graph classification, etc., can better demonstrate the effectiveness of the proposed method. This is especially important for a technical paper where new approaches are introduced, as opposed to the previous observation-based papers such as NLGraph and GPT4Graph.
>
> Firstly, I would like to emphasize that NLGraph[1] primarily serves as a benchmarking study. The NLGraph paper itself mentions that "we propose NLGraph (Natural Language Graph), a comprehensive benchmark of graph-based problem solving designed in natural language".  Given our paper's objective to evaluate and boost the graph reasoning abilities of large language models, we find it logical to design our tasks in a manner that closely mirrors the NLGraph benchmark.
>
> Additionally, it's important to note that the use of 'artificial' tasks for benchmarking the reasoning abilities of LLMs is a well-established practice in our field. An example of this is the GSM8K dataset[2], widely employed to evaluate mathematical reasoning skills. This particular dataset consists of 'simple' problems, all of which can be accurately solved using arithmetic operations at the middle school level.
>
> Somewhat paradoxically, the established node classification benchmarks like ogbn-arXiv (with textual feature) might pose lower demands on LLMs' graph reasoning abilities compared to the tasks presented in this text. For LLMs, determining the arXiv category of a paper based on its abstract is a very simple task. (LLMs can even continue writing the paper or provide review comments based on the abstract.) As a result, measuring the benefits of graph reasoning abilities for this task has become quite challenging (see Table 16 & 17 in [3]). In the era of LLMs, building a real-world benchmark that truly accurately measures the graph reasoning capabilities of these models requires the effort of the entire community. After careful consideration, at this early stage, we've opted for artificial datasets to secure somewhat precise measurements.
>
> > Q2 One major advantage of using LLMs compared to GNNs is explainability. Therefore, I would recommend conducting some analyses in this aspect.
>
> Thank you for your inquiry regarding the explainability of GraphLLM. In addressing the concern about providing explanations for the output answer, we conduct additional experiments on Shortest Path, tasked the LLM to simultaneously generate the optimal path as an explanation for the answer.
>
> Under the division of 6000/2000/2000 graph instances for training/validation/test, GraphLLM (LLaMA-7B-2) can output valid and optimal paths for 94.55% of the test graph instances while the LoRA fine-tuning of LLaMA-7B-2 achieved only 57.70%. This outcome provides preliminary evidence of GraphLLM's ability to articulate explanations for its predictions.
>
>
> [1] Can Language Models Solve Graph Problems in Natural Language? (NeurIPS 23) https://arxiv.org/abs/2305.10037
>
> [2] https://paperswithcode.com/dataset/gsm8k
>
> [3] Exploring the Potential of Large Language Models (LLMs) in Learning on Graphs https://arxiv.org/abs/2307.03393

---

> ### Author Response · Authors · 2023-11-19
> **Response to Reviewer DjB6 [2/3]**
>
> > Q3  I also wonder whether the authors have considered directly comparing with graph Transformers or GNNs, to see whether LLMs have advanced the field.
>
> We respectfully disagree and would like to stress that graph neural network-based methods are not a meaningful baseline for the issue explored in this paper.
>
> Firstly, our paper's primary aim is to enhance the graph reasoning abilities of LLMs, a goal that does not necessitate a comparison with non-LLM methods. Previous benchmarking work[1] has also not considered graph neural networks as a type of baseline. As corroborated by previous benchmarking work[1], graph neural networks have not been considered a standard baseline in this context. Our focus is not on outperforming graph neural network-based methods in graph reasoning tasks, such as shortest path problems, which can be precisely solved with simple algorithms with a few lines of code. Instead, the pivotal question we aim to explore is: in the trend where generative LLMs are increasingly seen as a universal agent due to their powerful reasoning capabilities, how these LLMs fare in graph reasoning. Considering this, the performance metrics of graph neural network-based methods do not contribute additional insights to our research question.
>
> Even if we were to attempt a comparison with graph neural networks, please note that the graph reasoning task we address (as outlined in Definition 2.2) is intrinsically a natural language generation task, necessitating inputs and outputs in human language. For instance, in a substructure counting task, the output is expected to be a natural language statement like 'There is 1 [substructure name]'. If we do not transform the task or adapt the graph neural network-based methods, even achieving a comparison becomes impossible. The question is how to make a fair comparison between a generative LLMs that take natural language for input and output, and a graph neural network?  Transforming natural language outputs into some kind of classification category seems a possible solution, but we should note that an inherent advantage of LLMs is their ability to interact and reason in human language, making such a comparison evidently meaningless.
>
> > Q4 the proposed method needs to be trained end-to-end, it increases the computation cost and can only operate with open-sourced LLMs (in order to obtain gradients), which limits the applicability.
>
> Imagine this application scenario: users requires an LLM (like LLaMA 2 7B) to address needs that can be abstracted as a shortest path problem. It is found that the outcomes of zero/few-shot prompting do not satisfy (Exact Match Accuracy: 0.1089). In such situations, the most common practice for users is to perform fine-tuning on this task. The point is, in this scenario, fine-tuning is a necessary cost. While the graph2text approach bears this cost, it does not yield satisfying outcomes. This, we argue, adequately showcases the superiority of our method in applicability.
>
> We wish to emphasize that our method indeed cannot be used with closed-source models, a limitation that should be attributed to the practice of closed-sourcing itself, rather than our method. We would like to remind that in many scenarios, using APIs from closed-source models is impractical, due to concerns such as data security or cost.
>
> > Q5  All things considered, the technical contribution of the paper is okay but not too novel, considering that all three components are heavily based on the existing literature, i.e., node understanding is a standard Transformer, structure understanding is essentially equivalent to Ma et al., 2023, and prefix tuning follows the common practice of LLMs. It would make the paper stronger if these components could be customized to further enhance the performance.
>
> We wish to emphasize that the paper's principal novelty lies in the idea of collaborating graph learning models and LLMs as a unified system, a first in this domain. Furthermore, even combining graph transformers with prefix tuning is non-trivial in its implementation. We would like to point out that we have indeed made customizations. For example, the most critical issue is how to make the graph transformer generate prefixes of specific dimensions. Our initial experimental attempts indicate that simply applying a dimensional transformation to the output of the graph transformer does not work. Therefore, the graph transformer's input is designed to be derived from a query of the same dimension as the prefix, a design we propose for the first time.
>
>
> [1] Can Language Models Solve Graph Problems in Natural Language? (NeurIPS 23) https://arxiv.org/abs/2305.10037

---

> ### Author Response · Authors · 2023-11-19
> **Response to Reviewer DjB6 [3/3]**
>
> > Q6 Minor: the authors claim one advantage of graph Transformers over GNNs is “its decoupling of node information and structural information”. Experimental evidence could be provided to support this claim.
>
> May I kindly direct your attention to Appendix C.2, entitled 'Ablation Study on Graph Transformer', where you will find the relevant results. The key distinction between GAT and Graph Transformer is that the former treats the graph structure as an inductive bias for aggregating node features, whereas the latter considers it as positional encodings.
>
> We appreciate your thoughtful feedback and welcome any additional questions or suggestions you may have.

---

### Official Review · Reviewer_B4te · 2023-11-01

**Soundness:** 3 good
**Presentation:** 3 good
**Contribution:** 3 good
**Rating:** 5
**Confidence:** 4

**Summary:**

GraphLLM introduces a new method for boosting graph reasoning abilities of LLMs. It introduces a graph encoder using a transformer encoder-decoder for encoding node text descriptions and a graph transformer for incorporating structure. These are combined to generate key value prefixes for an LLM to use.

They show results on 4 different small-size graph reasoning tasks. They show great performance on these tasks compared to the text only (graph2text) baselines. They also show a large reduction in the number of input tokens to the LLM.

**Strengths:**

This paper offers a clever approach to incorporating graph reasoning into LLMs.

1) The architecture is smart and sensible, and the integration via prefix tuning makes this relatively simple to integrate.

2) The experimental results are very strong compared to the baselines. GraphLLM essentially completely solves these tasks

3) The graph transformer design is important, as evidenced by the ablation in table 7.

Overall, the main strength of this paper is the architecture design. The architecture is quite smart. It can utilize node text features as well as  graph features to solve some tasks.

**Weaknesses:**

The main weaknesses of this paper are in its experimental evaluation. The architecture seems potentially powerful but under explored.

LLMs are general purpose reasoners over text and graph2text takes advantage of that. GraphLLM is purposely built and trained for these specific graph reasoning tasks. It would be shocking if it didn't do better. As a consequence, most of the results are not "interesting". Graph2text is a single representation that can be used for all four tasks. The prefixes from GraphLLM are tailored to each task individually. Yes, LLM fine-tuning is a more comparable setting, but also less interesting as once we are finetuning, we could be using many other architectures (e.g. GNN). The more interesting question would be can we use GraphLLM as a general graph->prefix encoder that enables the LLM to perform different fundamental tasks better

This leads to the second weakness, the experiments do not show that GraphLLM is "useful". The tasks tested do not require machine learning, they can all be solved explicitly. Why would I use a complex ML architecture to replace a simple program? I think the architecture is smart and there are many avenues to demonstrate utility but these experiments do not demonstrate it. The experiments do demonstrate that GraphLLM is capable of performing graph reasoning tasks and there is value there. There would be much more value in showing more general capabilities.

GPT-4 shows pretty good results on 3 of the tasks with few shot CoT. While still significantly below GraphLLM, LLMs alone may be able to learn these tasks if large enough and especially if fine-tuned or the prompt optimized. Would be interesting to see llama2-70b finetuned on these tasks.

The comparison of input tokens does not seem to account for the text input to the node understanding encoder. In fairness, this is a much smaller encoder so it does not matter as much.
However, the main issue here is that the design of graph2text seems intentionally designed to maximize the number of tokens. Example: For the max triplet sum task, why does it matter that "[Cornelia Brooks's] petite frame and delicate features give her a dainty and ethereal presence"? (Appedix D). In addition, the text uses the term "connected with" to describe edges but the question itself asks about "friends". I think this results in a poor showcasing of the LLM baselines. Even reading the shortest path text seems confusing as a human. Does the distance from earth matter? What does activating a wormhole mean? Why are we even dealing with "wormholes"? The only description of the task is "Starting from wormhole 1, How much dark matter we’ll need at the minimum to reach Wormhole 2?"

**Questions:**

Questions:

"The encoderdecoder is newly initialized and updated with the guidance of the pre-trained LLM." This is very confusing and it is not clear what it means? How does the pre-trained LLM provide guidance?
Is the encoder-decoder initialized from a pre-trained model or is it only the token embeddings?

Shouldn't prefix tuning require an embedding for each prefix value as well as each key? i.e. L x 2K x d?

What are the instructions for generating each node's text features? Why can't this just be done using a template?
Are the instructions generated for each node

For task 1, is it always the same substructure being identified?

For all the tasks is the input to the LLM (after the prefix) always the same?

How is exact match defined? Does it mean exact match appears in the output somewhere? Looking at the GPT-4 failure case examples in the appendix, it is hard to tell how this metric is applied?

Suggestions:
(Forgive me if any of these were done and I missed it)

Explicitly mention that in prefix tuning, the LLM parameters are fixed.

I don't think $d$ is explicitly defined. I take it as the dimension of the node and graph understanding encoder/decoders.

Stylistically, it looks better if G, E and V have a consistent style

In the introduction, I am not sure "inefficiencies" is the right word choice. What is inefficient about it? "difficulties"?

---

> ### Author Response · Authors · 2023-11-21
> **Response to Reviewer B4te [1/2]**
>
> Dear Reviewer B4te,
>
> Great thanks for your insightful comments! We really appreciate your feedback and here are our responses to some of your concerns.
>
> We must admit that our framework's ability to perform basic graphic tasks is indeed not very exciting. In evaluation, the more valuable aspect might be that finetuning on Graph2Text cannot achieve basic graph reasoning, such as the shortest path example. This suggests that, at least for now, the idea that 'natural language is all a graph needs' is still not true.
>
> We also attempted to use GraphLLM as a general framework to perform different fundamental tasks. When joint training Substructure Counting and Maximum Triplet Sum with the same setting in the paper, GraphLLM can simultaneously achieve accuracies of 0.9946±0.0021 on Substructure Counting and 0.8472±0.0210 on Maximum Triplet Sum, indicating its preliminary general capability.
>
> We fully agree that conducting experiments on more complex, real-world datasets would add significant value to this paper, and our considerations for using simple artificial tasks are as follows:
> - The use of simple artificial tasks for benchmarking the reasoning abilities of LLMs is a well-established practice in this field. An example of this is the GSM8K dataset[1], which is widely employed to evaluate LLM's mathematical reasoning abilities.
> - Somewhat paradoxically, the established real-world benchmarks like ogbn-arXiv (with textual feature) might pose lower demands on LLMs' graph reasoning abilities compared to the tasks presented in our paper. For LLMs, determining the arXiv category of a paper based on its abstract is a very simple task. (LLMs can even continue writing the paper or provide review comments based on the abstract.) As a result, measuring the benefits of graph reasoning abilities for this task has become quite challenging (see Table 16 & 17 in [2]). In the era of LLMs, building a real-world benchmark that truly accurately measures the graph reasoning capabilities of these models requires the effort of the entire community. After careful consideration, at this early stage, we've opted for artificial datasets to secure somewhat precise measurements.
>
> We are currently trying to further apply our architecture to some more exciting real-world applications.
>
> Also thanks for the concerns regarding the Graph2Text prompts. In Maximum Triplet Sum, the node text feature includes the key information about the person's age, and other descriptions generated by GPT-3.5-Turbo. This design aims to assess the node understanding capabilities of both GraphLLM and Graph2Text-based LLMs, specifically their ability to extract essential details from seemingly unrelated text. And according to your other concerns, we have revised the prompt and assessed Graph2Text with GPT-4 few-shot CoT. In Maximum Triplet Sum, we describe the edge between people using "friendship", and the accuracy remains 93.33%. In Shortest Path, we modify the background to cities connected with roads, where it incurs a cost to pass through a city. However, the accuracy decreases from 86.67% to 83.33%.
>
> [1] https://paperswithcode.com/dataset/gsm8k
>
> [2] Exploring the Potential of Large Language Models (LLMs) in Learning on Graphs https://arxiv.org/abs/2307.03393

---

> ### Author Response · Authors · 2023-11-21
> **Response to Reviewer B4te [2/2]**
>
> Thank you for your meticulous questions and the following are our responses to each question.
>
> > Q1 "The encoderdecoder is newly initialized and updated with the guidance of the pre-trained LLM." This is very confusing and it is not clear what it means? How does the pre-trained LLM provide guidance? Is the encoder-decoder initialized from a pre-trained model or is it only the token embeddings?
>
> We sincerely appreciate your feedback, and regret any confusion caused by the sentence. In the node understanding encoder-decoder, the nodes' textual descriptions are embedded using the frozen embedding table from the pre-trained LLM, while the encoder-decoder itself is randomly initialized. The phrase "under the guidance of the pre-trained LLM" is employed because during end-to-end training, the encoder-decoder's gradient is backpropagated from the LLM, allowing the encoder-decoder and the graph transformer to learn alignment with the pre-trained LLM.
>
> > Q2 Shouldn't prefix tuning require an embedding for each prefix value as well as each key? i.e. L x 2K x d?
>
> Thank you for your insightful query. Regarding the difference between our implementation and the original prefix-tuning, we employ a single graph-enhanced prefix token $p \in \mathrm{R}^{d^\mathrm{M}}$ that undergoes linear transformation by $W_V$ and $W_K$ to obtain the prefix value and prefix key. This approach only requires a prefix tensor of shape $L \times K \times d^{\mathrm{M}}$, allowing GraphLLM to save trainable parameters while maintaining effectiveness.
>
> > Q3 What are the instructions for generating each node's text features? Why can't this just be done using a template? Are the instructions generated for each node?
>
> We didn't use a fixed template to generate the nodes' text feature in all tasks. In Maximum Triplet Sum, we insert the key information randomly to the nodes' text, preventing that the model solves the task relying on some correlations instead of understanding and extracting the key information. For example, if the key information appears in a fixed position in the text, the node understanding transformer may recognize the specific positional embedding and extract the information from the position rather than understanding it. In the meanwhile, non-template text is considered a more realistic setting. So in Maximum Triplet Sum, we randomly select different sentences (generated by GPT-3.5-Turbo) describing a person and randomly insert a sentence containing the age to form each node's text feature.
>
> > Q4 For task 1, is it always the same substructure being identified?
>
> In the paper, it's always the C-C-O triangles being identified. And we conducted additional experiments to have GraphLLM simultaneously identify C-C-O triangles as well as C-O-O triangles. After joint training, GraphLLM achieves accuracies of 0.9925±0.0046 on C-C-O triangles and 0.9996±0.0001 on C-O-O triangles.
>
> > Q5 For all the tasks, is the input to the LLM (after the prefix) always the same?
>
> Yes, for the same task, the input instruction to the LLM is always the same.
>
> > Q6 How is exact match defined? Does it mean exact match appears in the output somewhere? Looking at the GPT-4 failure case examples in the appendix, it is hard to tell how this metric is applied?
>
> In both fine-tuning and in context learning prompting settings, LLM will output the answer in a desired format. So exact match compares the answer part of the LLM's output with the label to check correctness. In GPT's zero-shot setting (as shown in the appendix), we manually check each output and record the accuracy.
>
> We sincerely appreciate your suggestions and have revised the paper accordingly. The following are our responses to some of the questions in your suggestions part.
>
> > I don't think $d$ is explicitly defined. I take it as the dimension of the node and graph understanding encoder/decoders.
>
> We acknowledge the oversight in the definition of $d$. It indeed represents the dimension of the node understanding encoder-decoder and the structure understanding graph transformer.
>
> > In the introduction, I am not sure "inefficiencies" is the right word choice. What is inefficient about it? "difficulties"?
>
> We appreciate your scrutiny of the term 'inefficiencies' in the introduction. We acknowledge that 'difficulties' is a more accurate word choice to convey the challenges faced by LLM in processing sequential graph descriptions.
>
> Thank you again for your insightful comments. We look forward to further discussions with you.

---

### Official Review · Reviewer_ryJZ · 2023-11-02

**Soundness:** 3 good
**Presentation:** 3 good
**Contribution:** 2 fair
**Rating:** 5
**Confidence:** 4

**Summary:**

The paper proposes GraphLLM, a new approach to enhance the ability of large language models (LLMs) to understand and reason about graph data. It identifies a key limitation of current methods that convert graphs to text (Graph2Text), which forces LLMs to implicitly learn graph structures and results in lengthy contexts. GraphLLM integrates a graph learning module with the LLM via end-to-end training. This allows the LLM to leverage the graph module's strengths in an efficient way through a condensed graph-enhanced prefix. Experiments on 4 graph reasoning tasks show GraphLLM substantially improves accuracy over Graph2Text methods while reducing context length and accelerating inference.

**Strengths:**

* Novel integration of graphs and LLMs.

GraphLLM proposes a novel end-to-end approach to combine graph learning models and LLMs. This allows each component to be optimized to complement the other for different graph reasoning tasks.

* Significant performance gains.

GraphLLM improves accuracy by 54.44% on average over the best Graph2Text method, showing its effectiveness. Also, GraphLLM reduces context length by 96.45% compared to Graph2Text, enhancing efficiency. Besides, GraphLLM achieves 3.42x faster inference over the best Graph2Text method due to context reduction.

**Weaknesses:**

* Limited graph tasks evaluated.

The paper evaluates GraphLLM on four graph reasoning tasks, including substructure counting, maximum triplet sum, shortest path, and bipartite graph matching. Although these tasks cover basic graph reasoning abilities, they are still relatively simple, which might overestimate how GraphLLM performs on noisier graphs with complex relational patterns. More complex graph reasoning tasks could better demonstrate the capabilities and limitations of GraphLLM. For example, tasks requiring multi-hop reasoning on large graphs might be more challenging.

* Lack of analysis on scalability.

The datasets used in the paper are all small with fewer than 20 nodes on average. Thus, the ability of GraphLLM to handle large, real-world graphs is unclear. Evaluations on larger datasets would be informative.

* Restricted node features.

The tasks only use textual node features. Testing on graphs with non-textual node attributes would make the approach more broadly applicable.

* No comparison to other graph models.

The paper only compares GraphLLM against methods that convert the graph to text (Graph2Text). However, established graph neural networks like GNNs, GCNs, and graph transformers have been optimized specifically for graph reasoning. Comparing to these dedicated graph reasoning models could better highlight if GraphLLM actually improves over the state-of-the-art in graph representation learning. For example, how does GraphLLM compare to a graph transformer without the attached LLM on the tested graph reasoning tasks? This could isolate the benefits of the LLM integration. Similarly, comparisons to GNN variants on standard benchmarks could reveal where GraphLLM excels or lags behind existing graph-specific architectures. The improvements over Graph2Text may simply be because LLMs struggle with learning from text descriptions of graphs. Graph models designed for relational reasoning may be more competitive. In essence, without comparing to specialized graph reasoning models, it is hard to discern if GraphLLM's integration of LLMs and graph networks is truly advancing state-of-the-art performance. Adding these comparisons would give a clearer sense of GraphLLM's capabilities relative to the field of graph representation learning overall.

**Questions:**

Please check the weakness section above.

---

> ### Author Response · Authors · 2023-11-12
> **Response to Reviewer ryJZ**
>
> Dear reviewer ryJZ,
>
> Thank you for your review and valuable comments. We have carefully considered your feedback and provide here our detailed responses to each point.
>
> > Q1 Limited graph tasks evaluated.
>
> We wish to emphasize that the difficulty of our task is similar to that of the tasks designed in the benchmark study[1] on graph reasoning abilities of LLMs, which was recently accepted by NeurIPS 23.
>
> A primary aim of this paper is to attempt to answer the question: What hinders the graph reasoning abilities of LLMs? A natural thought is that, to investigate this issue, we should start with the most basic graph reasoning tasks. Surprisingly, we find that current LLMs even struggle with these simple tasks.  We agree that for further applications, dealing with noise graphs and more complex graph reasoning is important. However, for this article, we want to first focus on the primary question.
>
> > Q2 Lack of analysis on scalability.
>
> Thank you for your insightful observation regarding the scalability of large graphs. In our study, inspired by the previous work[1], we intentionally mirrored the graph size of ~20 in the graph reasoning problems from NLGBench to maintain consistency and focus. Surprisingly, LLMs, known for their general reasoning abilities, struggle with these basic graph reasoning tasks. Our primary goal is to explore and potentially resolve this apparent paradox. We fully acknowledge and agree with your point on the importance of handling larger graphs for practical applications. However, such an extension, while valuable, falls outside the scope of our current research focus.  We will include a discussion on scalability in the revised version of the paper.
>
> > Q3 Restricted node features.
>
> What we want to clarify is that, rather than designing models applicable to various real-world graphs, the main purpose of this paper is to investigate the reasons hindering the graph reasoning abilities of LLMs and to identify potential solutions. Non-textual node features requiring additional alignment with LLMs could cause unnecessary distractions in our investigation.
>
> > Q4 No comparison to other graph models.
>
> We respectfully disagree and would like to stress that graph neural network-based methods are not a meaningful baseline for the issue explored in this paper.
>
> Firstly, our paper's primary aim is to enhance the graph reasoning abilities of LLMs, a goal that does not necessitate a comparison with non-LLM methods. Previous benchmarking work[1] has also not considered graph neural networks as a type of baseline. As corroborated by previous benchmarking work[1], graph neural networks have not been considered a standard baseline in this context. Our focus is not on outperforming graph neural network-based methods in graph reasoning tasks, such as shortest path problems, which can be precisely solved with simple algorithms with a few lines of code. Instead, the pivotal question we aim to explore is: in the trend where generative LLMs are increasingly seen as a universal agent due to their powerful reasoning capabilities, how does the graph reasoning ability of LLMs perform, being one of the key reasoning skills for solving many problems like path planning, and towards what future developments might we be moving? Considering this, the performance metrics of graph neural network-based methods do not contribute additional insights to our research question.
>
> Even if we were to attempt a comparison with graph neural networks, please note that the graph reasoning task we address (as outlined in Definition 2.2) is intrinsically a natural language generation task, necessitating inputs and outputs in human language. For instance, in a substructure counting task, the expected output is a natural language statement like 'There is 1 [substructure name]'. If we do not transform the task or adapt the graph neural network-based methods, even achieving a comparison becomes impossible. The essential question is how to make a fair comparison between generative LLMs that take natural language for input and output, and a graph neural network?  Transforming natural language outputs into some kind of classification category seems a possible solution, but we should note that an inherent advantage of LLMs is their ability to interact and reason in human language, making such a comparison meaningless.
>
> [1] Can Language Models Solve Graph Problems in Natural Language? (NeurIPS 23)  https://arxiv.org/abs/2305.10037

---

### Official Review · Reviewer_wWow · 2023-11-06

**Soundness:** 2 fair
**Presentation:** 3 good
**Contribution:** 2 fair
**Rating:** 3
**Confidence:** 4

**Summary:**

This paper introduces GraphLLM, an approach to integrate graph learning models with LLMs via graph transformer and prefix tuning. The authors compare with baselines across four graph reasoning tasks.

**Strengths:**

I can see the contribution of this paper in trying to apply LLMs for graph learning/reasoning tasks, which I think is interesting.

The paper is well-organized.

Other than the performance comparison, the authors report the efficiency comparison, which I appreciate.

The authors further report the performance of gpt-3.5-turbo and gpt-4, other than LLama.

**Weaknesses:**

The authors claim their proposed method is an end-to-end approach. I wonder the applicability of their model to existing pre-trained LLMs. Can the proposed method be easily adapted or integrated into existing pre-trained LLMs without fine-tuning? If so, what is the formal definition of end-to-end? If not, will this introduce a large burden for training resources? It seems to me that end-to-end is not a good strategy for LLMs, which contradicts the most powerful capability of one model to fit multiple tasks and applications in LLMs.

There are multiple ways to condense the length and size of context windows, especially in the community of natural language processing. The authors should be more cautious about this when presenting this as the contribution, and should consider comparing their methods with baselines and approaches that address the length and size constraints in the NLP community.

Ambiguous preliminary. In the paper of prefix tuning, the prefix indicates a vector that concats to the front of the input and the representations in hidden layers. In the prefix tuning section of this paper, the authors introduce notations of QKV while not giving formal definitions that correspond to the graph learning scenario. Moreover, what is the input here, only K and V? Should the input also be Q? Before each layer in the transformer, where should the prefix be concatenated? How is this related to the graph learning scenario?

Limited technical novelty. This paper simply combines the graph transformer and prefix tuning of LLMs. In particular, the graph transformer serves as a learning model on top of the graph, then the output of the graph transformer is concatenated to LLMs via prefix tuning. The technics of graph transformer and prefix tuning are both well built, studied and employed by the research community.

Have authors consider comparing with or integrating P-tuning v2, which has shown better performance than prefix tuning.

I suspect that different instructions can cause the LLMs to generate significantly different results and performance. Have the authors conducted experiments or analyses to investigate different instructions? How did the authors choose the instructions?

All the tested graphs are relatively small. I wonder if the proposed method can be used in large-scale graphs such as ogb-arxiv/ogb-products/ogb-ppa. If the proposed method can be extended to large graphs, how will the authors sample the neighbors? Will neighbor sampling affect the performance?

Lack of baselines. One important line of baselines that are missed here is the graph neural network-based ones for graph reasoning tasks. I don’t think the experiments are comprehensive without comparing to methods that are designed to solve the graph reasoning problem. In fact, it is surprising that the authors only compare with NLP-based methods on solving graph tasks.

**Questions:**

see above.

---

> ### Author Response · Authors · 2023-11-19
> **Response to Reviewer wWow [1/3]**
>
> Dear reviewer wWow,
>
> Thank you for your review and valuable comments. We have carefully considered your feedback and provide here our detailed responses to each point.
>
> > Q1-1 I wonder the applicability of their model to existing pre-trained LLMs. Can the proposed method be easily adapted or integrated into existing pre-trained LLMs without fine-tuning?
>
> Imagine this application scenario: users requires an LLM (like LLaMA 2 7B) to address needs that can be abstracted as a shortest path problem. It is found that the outcomes of zero/few-shot prompting do not satisfy (Exact Match Accuracy: 0.1089). In such situations, the most common practice for users is to perform fine-tuning on this task. The point is, in this scenario, fine-tuning is a necessary cost.
>
> > Q1-2 It seems to me that end-to-end is not a good strategy for LLMs, which contradicts the most powerful capability of one model to fit multiple tasks and applications in LLMs.
>
> We acknowledge the recognized strength of leveraging a single model to address diverse tasks in LLMs. However, our experimentation reveals that prompting open-source LLMs and GPT-3.5-turbo for these tasks yields unsatisfying accuracy, rendering it impractical to rely on a single model for solving multiple problems. Even fine-tuning the LLM for a specific downstream task fails to yield satisfactory performance.
>
> In contrast, GraphLLM exhibits the ability to effectively address individual downstream tasks. Our additional experiments demonstrate that GraphLLM possesses a preliminary capacity to handle multiple tasks concurrently. For instance, joint training of GraphLLM on Substructure Counting and Maximum Triplet Sum results in an accuracy of 0.9946±0.0021 on Substructure Counting and 0.8472±0.0210 on Maximum Triplet Sum. Similarly, when training GraphLLM to count the number of C-C-O triangles and C-O-O triangles in Substructure Counting, it achieves simultaneous accuracies of 0.9925±0.0046 on C-C-O triangles and 0.9996±0.0001 on C-O-O triangles.
>
> > Q2 Baselines about condensed prompt size.
>
> We acknowledge the effectiveness of compression techniques such as Selective Context[1] in shortening prompt length and enhancing the efficiency of Graph2Text methods. However, it's important to highlight the substantial differences between Graph2Text and GraphLLM. Even when filtering 50% of the context in Graph2Text, the remaining 50% is still much longer than GraphLLM's instruction. GraphLLM achieves a 96.45% reduction in context length compared to the original context, and when half of the original tokens are filtered, the reduction is still 92.9%. It's worth noting that the filtering process may result in the removal of key information related to node details or the graph structure, potentially leading to additional accuracy decrease in the baseline.
>
> > Q3-1 In the prefix tuning section of this paper, the authors introduce notations of QKV while not giving formal definitions that correspond to the graph learning scenario.
>
> In the preliminary, we present the prefix tuning without involving graph learning scenario. Specifically, $ Q_l, K_l, V_l $ denote the query, key, and value matrices generated by the input tokens of the $l$-th transformer layer. Note that the input to the LLM transformer is solely the "instruction" and is independent of graph-related elements.
>
> > Q3-2 Moreover, what is the input here, only K and V? Should the input also be Q? Before each layer in the transformer, where should the prefix be concatenated?
>
> In prefix tuning, only $K_l, V_l$ need to be prepended with the prefix tokens $P_l$; $Q_l$ remains unaffected. The prepended $K_l^{\prime}, V_l^{\prime}$ as well as the original $Q_l$ are collectively fed into the attention mechanism. Notably, the prefix tokens $P_l \in \mathrm{R}^{K \times d^{\mathrm{M}}}$ are prepended to the original keys and values $(K_l, V_l \in \mathrm{R}^{* \times d^{\mathrm{M}}})$ in the first dimension, resulting in $K_l^{\prime}, V_l^{\prime} \in \mathrm{R}^{(K+ * ) \times d^\mathrm{M}}$, where $K$ is the number of prefix tokens, $*$ denotes the length of the input instruction, and $d^\mathrm{M}$ is the dimension of the LLM.
>
> > Q3-3 How is this related to the graph learning scenario?
>
> We illustrate the relationship of prefix tuning and graph learning in Sec 3.4, "Graph-enhanced Prefix Tuning for LLMs". The prefix tokens are a combination of graph representations and learnt prefix embeddings: $P = G W_{\mathrm{U}} + B$, where $G$ is the graph representation from the graph learning module, $B$ is the learnt prefix embedding and $W_\mathrm{U}$ is a matrix converting dimension.
>
> [1] Compressing Context to Enhance Inference Efficiency of Large Language Models (EMNLP 23) https://arxiv.org/abs/2310.06201

---

> ### Author Response · Authors · 2023-11-19
> **Response to Reviewer wWow [2/3]**
>
> > Q4 Limited technical novelty.
>
> We beg to differ that the paper's principal novelty lies in the idea of collaborating graph learning models and LLMs as a unified system, a first in this domain. Furthermore, even combining graph transformers with prefix tuning is non-trivial in its implementation. For example, the most critical issue is how to make the graph transformer generate prefixes of specific dimensions. Our initial experimental attempts indicate that simply applying a dimensional transformation to the output of the graph transformer does not work. Therefore, the graph transformer's input is designed to be derived from a query of the same dimension as the prefix, a design we propose for the first time.
>
> > Q5 Have authors consider comparing with or integrating P-tuning v2, which has shown better performance than prefix tuning.
>
> We appreciate the opportunity to clarify a common misconception: P-tuning v2 is often perceived as a general enhancement over Prefix Tuning. The P-Tuning v2 paper itself mentions that "P-Tuning v2 is an implementation of [Prefix Tuning] optimized and adapted for NLU (Natural Language Understanding)". Originally, Prefix Tuning was designed for NLG (Natural Language Generation) purposes, while P-Tuning v2 adjusted this approach for NLU, employing classification heads for sequence labeling tasks instead of verbalizers. The impact of LLMs largely stems from their generative capabilities, and in the current landscape (as of 2023), most LLM-based research focuses on formulating tasks as NLG challenges. The use of classification heads post-LLM, a key differentiator between P-tuning v2 and traditional Prefix Tuning, is not a common approach in this domain.
>
> > Q6 I suspect that different instructions can cause the LLMs to generate significantly different results and performance. Have the authors conducted experiments or analyses to investigate different instructions? How did the authors choose the instructions?
>
> Thank you for the question regarding the impact of different instructions on language model performance. We acknowledge that varying instructions can influence results, but we posit that the magnitude of this influence may be limited. In the case of fine-tuning, such as through prefix-tuning, the learned prefixes effectively serve as optimized prompts for downstream tasks. This process helps mitigate the performance gap that may arise from different input instructions. In the case of prompting, the search for the optimal prompt remains an open question because it's not practical to experiment with all possible prompts. Nevertheless, we conducted experiments on Maximum Triplet Sum and Shortest Path, employing different instructions through GPT-4 few-shot CoT prompting. Specifically, our experiment with Maximum Triplet Sum remains an accuracy of 93.33% under the new instruction. And in the case of Shortest Path, we observed a decrease in accuracy from 86.67% to 83.33% with the new instruction.
>
> > Q7 All the tested graphs are relatively small. I wonder if the proposed method can be used in large-scale graphs such as ogb-arxiv/ogb-products/ogb-ppa. If the proposed method can be extended to large graphs, how will the authors sample the neighbors? Will neighbor sampling affect the performance?
>
> Thank you for your insightful observation regarding the processing of large graphs. In our study, inspired by the foundational work 'Can Language Models Solve Graph Problems in Natural Language? (NeurIPS 23)'[1], we intentionally mirrored the graph size of ~20 in the graph reasoning problems from NLGBench to maintain consistency and focus. Surprisingly, LLMs, known for their general reasoning abilities, struggle with these basic graph reasoning tasks. Our primary goal is to explore and potentially resolve this apparent paradox. We fully acknowledge and agree with your point on the importance of handling larger graphs for practical applications. However, such an extension, while valuable, falls outside the scope of our current research focus. We believe this area presents an exciting opportunity for future work.
>
>
> [1] Can Language Models Solve Graph Problems in Natural Language? (NeurIPS 23)
> https://arxiv.org/abs/2305.10037

---

> ### Author Response · Authors · 2023-11-19
> **Response to Reviewer wWow [3/3]**
>
> > Q8 Lack of baselines. One important line of baselines that are missed here is the graph neural network-based ones for graph reasoning tasks. I don’t think the experiments are comprehensive without comparing to methods that are designed to solve the graph reasoning problem. In fact, it is surprising that the authors only compare with NLP-based methods on solving graph tasks.
>
> We respectfully disagree and would like to stress that graph neural network-based methods are not a meaningful baseline for the issue explored in this paper.
> Firstly, the primary claim of this paper is to enhance the graph reasoning capabilities of Large Language Models (LLMs). To support this claim, there is no need to compare with methods outside of LLMs. Previous benchmarking work has also not considered graph neural networks as a type of baseline. The purpose of this paper has never been to achieve better performance with LLMs on certain graph reasoning tasks than graph neural networks/graph transformers; these tasks can be exactly solved with algorithms consisting of just a few lines of code. The paper also does not seek to use Large Language Models (LLM) for the improvement of graph representation learning.
> Moreover, the graph reasoning task we address (refer to Definition 2.2) is intrinsically a natural language generation task, requiring input and output in human language. For example, in the substructure counting task, the desired output is a natural language statement like 'There is 1 [substructure name]'. If we do not transform the task or adapt the graph neural network-based methods, even achieving a comparison becomes impossible. Adapting graph neural network-based methods to generate such natural language outputs presents a significant challenge, diverging from the scope of our investigation.
>
> We sincerely hope these clarifications address your concerns, and we remain open to further discussion or clarification.

---

### Meta-Review · Area_Chair_bscq · 2023-12-09

**Metareview:**

The paper proposes GraphLLM, an approach to integrate graph learning models with LLMs via graph transformer and prefix tuning. Specifically, the graph transformer serves as a learning model on top of the graph, then the output of the graph transformer is concatenated to LLMs via prefix tuning. Evaluation on four graph reasoning tasks shows advantage of the new method compared to existing generic methods like Graph2Text. However, reviewers have found that the evaluated graph tasks are limited, and the proposed method is finetuned on the target task, which makes it unimpressive even if the model outperforms other generic method that requires no finetuning. The technical novelty is somehow limited as both components (Graph transformer and prefix-tuning) are well-known.

**Justification For Why Not Higher Score:**

- limited evaluation tasks
- limited technical novelty
- unfair comparison

**Justification For Why Not Lower Score:**

- proposed new approach GraphLLM
- showed improvement on certain datasets

---

### Decision · Program_Chairs · 2024-01-16

Reject